# CACR: Reinforcing Temporal Answer Grounding in Instructional Video via Candidate-Aware Causal Reasoning

Muge Qi [1]   Rong Fu [2]   Pengbin Feng [3]   Xianda Li [4]   Yu Cai [1]   Yifu Guo [5]   Shizhe Zhang [1]   Simon James Fong [2]
Lei Ma[† 1]   Bin Li[† 6]

## Abstract

The task of temporal answer grounding in instructional video (TAGV), which aims to locate precise video segments that respond to natural language queries, is increasingly important for direct video answer retrieval. This task remains challenging due to the need to comprehend semantically complex questions and to address the significant length mismatch between untrimmed videos and short target moments. Existing methods often suffer from sensitivity to irrelevant content or insufficient visual reasoning capabilities. To tackle these limitations, we propose a Candidate-Aware Causal Reasoning (CACR) framework. Our approach first employs a Visual-Language Pre-training based Candidate Selection (VBCS) algorithm to efficiently generate K candidate segments, then applies a temporal logic reasoning module enhanced by a rejection reward mechanism and optimized via Group Relative Policy Optimization (GRPO) for robust inference. Extensive experiments on six benchmarks demonstrate that our method achieves state-of-the-art performance in terms of mean Intersection-over-Union (mIoU), providing a new perspective for reasoning-based retrieval in long videos.

## 1. Introduction

The rapid growth of video data has increased the demand for methods that can directly retrieve answers from videos in applications such as tutorials, fault diagnosis, and event reconstruction. In these scenarios, temporal continuity and relational dynamics are often more important than static images or text, making temporal answer grounding in instructional video (TAGV) a key research focus in AI and computer vision. Unlike traditional Visual Question Answering (VQA), which outputs text, or Temporal Sentence Grounding in Videos (TSGV), TAGV deals with semantically complex queries that often depend on visual demonstrations rather than pure language. This leads to a significant semantic gap and higher task complexity (Li et al., 2024b). An additional challenge in long videos is the extreme ratio between the total video length and the duration of the target answer segment. As shown in Figure 1A(a), the TutorialVQA(Colas et al., 2019) dataset exhibits an extreme ratio of 48.9—meaning the model must search through nearly 49 seconds of video to locate a 1-second answer. This highlights the urgent need for efficient long-video search mechanisms and fine-grained temporal localization.

Current approaches to TAGV suffer from several key limitations. Methods based on VLP models (Zhang et al., 2020a; 2021; Weng & Li, 2023; Li et al., 2024b) rely on low-level feature matching, making them sensitive to dynamic scenes and redundant content, and they often lack deeper relational reasoning. Self-supervised step segmentation methods (Dvornik et al., 2023; Tang et al., 2019) may miss subtle visual transitions and semantic relationships. Text-driven LLM-based approaches (OpenAI, 2023; Driess et al., 2023; Xiao et al., 2025) are constrained by subtitle quality and weak visual integration. While early RL-based LVLM methods (Yang et al., 2023; Guo et al., 2024; Dong et al., 2025; Zhao et al., 2025) suffered from overfitting and poor task adaptation, more recent approaches (Wang et al., 2025b;c) have shown promise by directly optimizing metrics such as IoU. However, they typically process the entire video without addressing the extreme answer-to-video duration ratio, leading to information loss and reasoning bias when the target segment is very short.

To tackle the core issue of extreme length disparity, we propose to narrow the processing scope for the subsequent reasoning model. We first employ a Visual-Language Pre-training based Candidate Selection (VBCS) algorithm to

†Corresponding authors. [1]National Biomedical Imaging Center, Peking University, Beijing, China [2]University of Macau, Macau, China [3]University of Southern California, Los Angeles, USA [4]University of Bologna, Bologna, Italy [5]Sun Yat-sen University, Guangzhou, China [6]Shenzhen Institute of Advanced Technology, Shenzhen, China. Correspondence to: Lei Ma <lei.ma@pku.edu.cn>, Bin Li <b.li2@siat.ac.cn>.

*Proceedings of the $43^{rd}$ International Conference on Machine Learning*, Seoul, South Korea. PMLR 306, 2026. Copyright 2026 by the author(s).

efficiently generate a manageable set of K high-quality candidate segments from the long video. This coarse screening focuses the model's attention on promising regions, alleviating information overload. Crucially, we observed that the maximum IoU between the Top-K candidate segments and the ground truth increases monotonically with K (Figure 1A(b,c)). This finding indicates that even if the single best candidate is imperfect, a larger, high-quality candidate pool is likely to contain a segment very close to the ground truth. Inspired by this, we introduce the Candidate-Aware Causal Reasoning (CACR) framework.

Within the refined candidate set, our framework employs a sophisticated temporal logic reasoning module. This module is strengthened by a rejection reward mechanism, encouraging the model to discard incorrect candidates when uncertain, thereby improving robustness. The reasoning process is optimized using Group Relative Policy Optimization (GRPO), which enhances policy learning by comparing groups of responses. To further aid reasoning, we integrate prior knowledge through an LLM-driven answer hypothesis generator and a subtitle summarization module, improving semantic alignment between the query and video content. Our main contributions are as follows.

(i) We propose the VBCS algorithm to efficiently generate high-quality candidate segments, mitigating the information overload in long videos.

(ii) We design a temporal logic reasoning module reinforced by a rejection reward mechanism and optimized via GRPO, enhancing distractor discrimination and causal reasoning capability. discrimination, enhances counterfactual robustness, steers the model away from label fitting, and promotes cross-scene generalization.

(iii) Extensive experiments on six benchmarks demonstrate that our method achieves state-of-the-art performance in mIoU, showcasing its strong generalization ability.

## 2. Related Work

**Temporal answer grounding in instructional Video (TAGV)** requires a model to localize the precise temporal segment in an untrimmed video that directly answers a natural language question. This task demands more sophisticated multimodal reasoning compared to Temporal Sentence Grounding in Videos (TSGV). Early approaches often followed a two-stage retrieve-and-verify paradigm (Zhang et al., 2020a;b), which first generated candidate segments and then performed fine-grained matching. Such methods, however, can be susceptible to dataset biases—such as the spurious correlation between moment location and prediction—that undermine generalization. To mitigate this, Yang

et al.(Yang et al., 2021) proposed a causality-inspired framework that explicitly models and removes the confounding effect of temporal location via causal intervention, thereby forcing the model to rely more on genuine visual content. Subsequent sliding-window Transformers (Lei et al., 2021a; Qu et al., 2020) framed TAGV as a set prediction task, directly regressing segment boundaries. In parallel, efforts to improve cross-modal matching have explored automated network design, such as the cross-modal neural architecture search method of Wang et al.(Yang et al., 2022), which searches for an optimal query-conditioned architecture to model complex video-text interactions. Methods built on the pre-train then fine-tune paradigm (Lei et al., 2021b; Yan et al., 2022) leveraged large-scale image-text pre-training before adapting to the task. While these methods have improved benchmark performance, they suffer from several limitations, including shallow cross-modal interaction, reliance on predefined windows or large-scale image-text pairs, and a fundamental misalignment with TAGV's requirement for precise temporal localization—often resulting in temporal-semantic mismatches. More recent caption-based approaches, such as VTPSL (Li et al., 2024b) and MutualSL (Weng & Li, 2023), enhance cross-modal alignment using video subtitles and pre-trained language models. However, they still lack the ability to model inter-step relationships and remain sensitive to redundant content (e.g., presenter irrelevant narration), leading to localization bias.

**Reasoning in Large Vision-Language Models(LVLMs)**. Recent studies have increasingly adopted end-to-end frame-based methods that fine-tune LVLMs through supervised fine-tuning (SFT) with autoregressive losses (Ren et al., 2024; Zeng et al., 2025; Hannan et al., 2025; Wu et al., 2025; Dong et al., 2025; Yang et al., 2023; Zhao et al., 2025). However, these methods often underperform compared to feature-based approaches. We argue that a key reason for LVLMs' suboptimal performance in temporal grounding lies in the excessive penalty imposed on false negatives during SFT. For example, in an action localization task where the ground truth spans frames 10–30, a model predicting frames 9–29 may still capture the essential action with only minor boundary shifts—well within human perceptual tolerance. Yet, autoregressive loss heavily penalizes such reasonable predictions, leading to overfitting and limited generalization. Inspired by recent breakthroughs in reinforcement learning (RL) for post-training large language models—such as GPT-4o1 (OpenAI, 2023), DeepSeek-R1 (Guo et al., 2025), Kimi-K1.5 (Team et al., 2025), Qwen-3 (Yang et al., 2025), and Magistral (Mistral-AI et al., 2025)—where methods like GRPO, RLVR, and Expert Iteration have advanced the state-of-the-art in code and mathematical reasoning, and encouraged by their recent extension to long-video reasoning in GRPO-enhanced multimodal LLMs (Zhang et al., 2025), we explore whether GRPO can serve as an effective

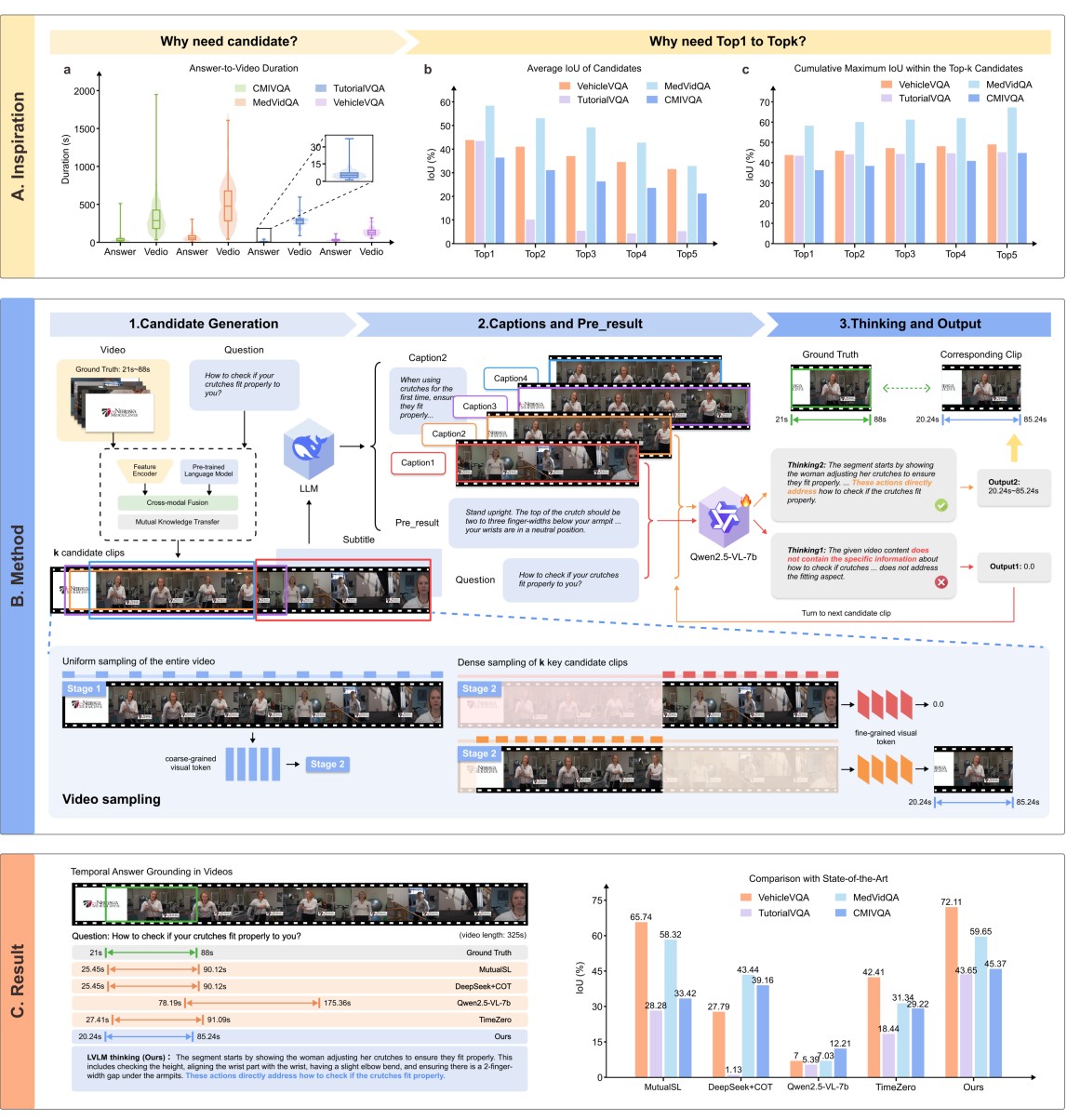

*Figure 1.* Overview of the proposed CACR framework. Fig. 1A illustrates the inspiration, showing the extreme length contrast between answer and video segments (a), the average IoU of candidates at different ranks (b), and the cumulative maximum IoU within Top-K selections (c). Fig. 1B outlines the method pipeline, including candidate generation (B.1), caption(Subtitle†) and answer hypothesis extraction (B.2), and causal reasoning for final prediction (B.3). Fig. 1C compares the performance of our approach with state-of-the-art methods.

alternative for TAGV.

**Group Relative Policy Optimization** is an emerging reinforcement learning technique that optimizes policy gradients by computing relative advantages within sample groups, thereby substantially reducing training variance and enhancing decision quality. Initially successful in text-based domains such as mathematics and programming (OpenAI, 2023; Guo et al., 2025), GRPO has demonstrated strong reasoning capabilities, enabling large models to approach human-level performance on complex tasks (Evstafev, 2025; OpenAI, 2023). Recent studies have extended GRPO to video structure understanding. Works such as (Ge et al., 2025; Wang et al., 2025a;b; Dong et al., 2025; Wang et al., 2025c) show that reinforcement learning-based post-training can significantly improve model reasoning in video-related tasks. However, these methods typically process the entire video without accounting for the extreme timespan ratio be-

tween the answer segment and the full video duration, often leading to substantial information loss and biased results.

# 3. Method

The TAGV task aims to temporally locate the video segment corresponding to a given text query in a long video. A video is represented as a sequence of frames $v_1, \ldots, v_T$, the language query is $Q$, and the target segment is defined by its temporal boundaries $[t_s, t_e]$ ($t_s, t_e \in \mathbb{R}^+$ in seconds).

We introduce CACR, a framework designed to enhance the capability of LVLMs for the TAGV task using a reinforcement learning approach. Unlike traditional RL pipelines that require a supervised fine-tuning (SFT) phase for cold-start initialization, CACR directly employs Group Relative Policy Optimization (GRPO) without explicit pre-training. This design choice is motivated by the strong inherent capabilities of our base model, Qwen2.5-VL-7B-Instruct, which already possesses robust instruction-following and reasoning abilities. The model's pre-existing proficiency in understanding temporal concepts and following structured output formats provides a sufficient foundation for effective policy optimization through reward signals alone. First, in Section 3.1, we introduce the VBCS framework to generate high quality candidate segments. Then, in Section 3.2, we describe the reinforcement learning training mechanism of CACR.

## 3.1. Top-K Candidate Generation via VBCS

To address the length disparity between answer segments and full videos, we leverage the Visual-Language Pre-training based Candidate Selection (VBCS) to generate high-quality top-$K$ candidate segments. VBCS is designed as a flexible framework that can incorporate any state-of-the-art VLP based localizer. Here we instantiate VBCS with the MutualSL (Weng & Li, 2023), a strong open-source temporal grounding model, to ensure precise cross-modal alignment.

Given an untrimmed video $V = \{v_i\}_{i=1}^T$, subtitle sequence $S = \{(t_j^{\text{start}}, t_j^{\text{end}}, s_j)\}_{j=1}^m$ (each tuple contains start/end times in seconds and text content), and query $Q$, VBCS outputs $K$ candidate segments $C_{\text{vis}} = \{(t_k^s, t_k^e)\}_{k=1}^K$ (in seconds) that aim to cover the ground-truth segment $[V_s^*, V_e^*]$ (in seconds). The workflow of VBCS (instantiated with MutualSL (Weng & Li, 2023)) is formalized in Algorithm 1.

The VBCS framework, instantiated with MutualSL (Weng & Li, 2023), operates through four sequential stages (Algorithm 1): (1) cross-modal feature extraction and fusion, (2) dual-predictor probability estimation, (3) cross-modal alignment through mutual knowledge transfer, and (4) temporally-extended candidate generation and selection. This approach addresses the length disparity challenge by narrowing the processing scope for subsequent modules

---

**Algorithm 1** Top-$K$ Candidate Generation via VBCS

---

**Require:** Untrimmed video $V$, subtitles $S$, query $Q$, candidate number $K$, temporal extension $\Delta t$, video duration T

**Ensure:** Top-$K$ candidate segments $C_{\text{vis}} = \{(c_k^s, c_k^e)\}_{k=1}^K$

1: **Stage 1: Cross-Modal Feature Extraction**
2: Extract visual features: $\mathbf{V} = \text{I3D}(V)$
3: Extract textual features: $\mathbf{T} = \text{PLM}([Q; S])$
4: Fuse features using Context-Query Attention
5: **Stage 2: Dual-Predictor Probability Estimation**
6: Visual predictor: estimate frame-level start/end probabilities $V_s, V_e$
7: Textual predictor: estimate token-level start/end probabilities $T_s, T_e$
8: **Stage 3: Cross-Modal Alignment (Training)**
9: Align predictions using timeline mapping $\mathbb{Q}(\cdot)$
10: Train with combined loss: base prediction loss + mutual transfer loss
11: **Stage 4: Top-$K$ Candidate Selection (Inference)**
12: Generate candidate segments by pairing top-ranked start/end positions: Candidates = $\{(\max(0, v_s - \Delta t), \min(v_e + \Delta t, T)) \mid v_s \in \text{Top-}K(V_s), v_e \in \text{Top-}K(V_e), v_s < v_e\}$
13: Select first $K$ candidates in generation order
14: **return** $C_{\text{vis}}$

---

while maintaining a high recall of the ground-truth segments. Details of MutualSL are deferred to Appendix A.2.

## 3.2. GRPO-based Temporal Localization Framework

After obtaining the top-$K$ candidate segments $C_{\text{vis}} = \{(t_{s_k}, t_{e_k})\}_{k=1}^K$ generated by VBCS, we propose **CACR** —a reinforcement learning framework based on GRPO. This framework drives LVLMs to perform deep reasoning over the candidate segments, rather than directly regressing timestamps. Specifically, the model adopts a "reason-first, localize-later" strategy, fully leveraging causal priors and text summaries generated by an LLM to enhance the understanding of step-level logic and semantic context. Furthermore, the reasoning process is optimized via a composite reward function that integrates temporal IoU reward, rejection reward, and template reward, improving the accuracy of temporal boundary prediction and robustness against distracting segments. This approach significantly mitigates localization bias and enhances generalization capability weakened by the lack of causal reasoning.

### 3.2.1. GRPO ALGORITHM FOUNDATION

As the core algorithm of our CACR framework, GRPO optimizes the policy model $\pi_\theta$ (i.e., the decision-making policy of the LVLM) through a rule-based reward function.

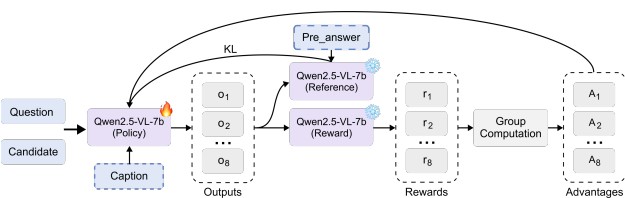

*Figure 2.* The policy model (Qwen2.5-VL-7B) takes the Question, Candidate segment, and Caption as input to generate multiple outputs ($o_1$–$o_8$). These outputs, along with a Pre-answer, are fed into the reward and reference models to compute rewards ($r_1$–$r_8$), which are then used to calculate advantages ($A_1$–$A_8$). The policy model is subsequently optimized using a KL divergence penalty.

It is particularly suitable for tasks with well-defined output spaces, such as temporal localization. The core mechanism operates as follows: Given an input question $Q$, the model generates $G$ candidate responses $\mathbf{o} = \{o_1, \ldots, o_G\}$. A designed reward function $r(\cdot)$ computes the rewards $\{r(o_1), \ldots, r(o_G)\}$. The objective of GRPO is to maximize the weighted sum of rewards:

$$R(\mathbf{o}) = \sum_{i=1}^{G} \frac{\pi_\theta(o_i)}{\pi_{\theta_{\text{old}}}(o_i)} \cdot \frac{r(o_i) - \mu_\mathcal{R}}{\sigma_\mathcal{R}}$$

where $\mu_\mathcal{R} = \text{mean}\left(\{r(o_i)\}_{i=1}^{G}\right)$ is the group mean of the rewards, $\sigma_\mathcal{R} = \text{std}\left(\{r(o_i)\}_{i=1}^{G}\right)$ is the group standard deviation of the rewards, $\pi_{\theta_{\text{old}}}$ denotes the policy parameters before the latest update. The normalization term $\frac{r(o_i) - \mu_\mathcal{R}}{\sigma_\mathcal{R}}$ serves as an estimate of the relative advantage of response $o_i$, denoted as $A(o_i)$. A positive $A(o_i)$ indicates that the response's reward is above the group average, while a negative value signifies below-average performance.

### 3.2.2. POLICY OPTIMIZATION OBJECTIVE

As illustrated in Figure 2, the policy optimization objective of GRPO aims to maximize the cumulative reward while constraining the currently trainable policy model $\pi_\theta$ via a KL divergence regularization term, preventing it from deviating excessively from a stable reference model $\pi_{\text{ref}}$. The objective function is defined as follows:

$$\max_{\pi_\theta} \mathbb{E} \left[ \sum_{i=1}^{G} \frac{\pi_\theta(o_i)}{\pi_{\theta_{\text{old}}}(o_i)} A(o_i) \right] - \beta \cdot D_{\text{KL}}(\pi_\theta \parallel \pi_{\text{ref}})$$

where: $\beta > 0$ is a regularization coefficient that balances reward optimization and policy conservatism, $A(o_i) = \frac{r(o_i) - \mu_\mathcal{R}}{\sigma_\mathcal{R}}$ denotes the relative advantage of response $o_i$ within the group, $D_{\text{KL}}(\pi_\theta \parallel \pi_{\text{ref}})$ measures the distributional divergence between the current policy and the reference model.

To address this issue, we propose a differentiated prompting strategy: The policy model $\pi_\theta$ receives the raw input state $s = (C_{\text{vis}}, Q, C_{\text{vis}_{\text{Subtitle}^\dagger}})$, where $C_{\text{vis}}$ denotes the visual features of the candidate segments, $Q$ is the user query, $C_{\text{vis}_{\text{Subtitle}^\dagger}}$ is an LLM-generated summary of the subtitles within those candidate segments.

The reference model $\pi_{\text{ref}}$ receives a semantically enhanced and reformulated input state $s' = (C_{\text{vis}}, Q, \text{Pre-answer})$, prompted with a statement like "Consider the following hypothetical answer: {Pre-answer}". Here Pre-answer is generated by LLM and represents a hypothetical answer to the query. This design provides the reference model with a higher-level semantic context focused on the query's intent.

This design ensures that the KL divergence term $D_{\text{KL}}(\pi_\theta \parallel \pi_{\text{ref}})$ acts not merely as a constraint against an initial policy, but rather guides the policy model towards outputs that are aligned with a more stable and intent-aware prior distribution. This mechanism significantly improves training stability and ensures that while pursuing high rewards, the model's predictions remain strongly correlated with the true semantic intent.

In the temporal localization task, the reward $r(o_i)$ is defined as the composite reward $R_{\text{total}}$ described in Section 3.2.3. The policy $\pi_\theta$ is optimized via gradient ascent, progressively favoring the generation of temporal boundaries that exhibit both high reward and strong semantic consistency.

### 3.2.3. COMPOSITE REWARD FUNCTION

The reward function consists of three components:

$$R_{\text{total}} = R_{\text{fmt}} + (1 - \alpha) \cdot R_{\text{IoU}} + \alpha \cdot R_{\text{rej}} \qquad (1)$$

The balance coefficient $\alpha$ is empirically set to 0.8 during training, emphasizing the importance of correct rejection decisions while maintaining strong temporal alignment incentives.

**Template Reward** $R_{\text{fmt}}$ encourages the model to follow structured output format: 1 when format is correct, 0 otherwise.

**Temporal Alignment Reward** $R_{\text{IoU}}$ activates whenever the output $(t_s^*, t_e^*)$ is not the rejection token $(0, 0)$ and satisfies $0 \leq t_s^* < t_e^*$

$$R_{\text{IoU}} = \frac{|[t_s^*, t_e^*] \cap [t_s^{\text{GT}}, t_e^{\text{GT}}]|}{|[t_s^*, t_e^*] \cup [t_s^{\text{GT}}, t_e^{\text{GT}}]|} \qquad (2)$$

**Rejection Reward** $R_{\text{rej}}$ equals 1 when the model outputs $[0.0, 0.0]$ and the candidate segment $C_k$ has no overlap with ground truth (IoU=0), otherwise 0.

### 3.3. Inference Pipeline

During inference, the pipeline processes an input video through three main stages to produce the final temporal

prediction. The specific steps are as follows:

**Stage 1: Candidate Proposal Generation** The input video is processed by the VBCS, which predicts $K$ candidate segments based on an analysis of video content features. These segments represent temporal intervals likely to contain relevant information, forming a preliminary set of temporal proposals.

**Stage 2: Subtitle[†] and Pre-Answer Generation** This stage produces *two* semantic aids of *different scope* via two independent LLM calls (full prompts in Appendix A.7.2):

- **(a) Pre-answer (per-query, candidate-agnostic).** A single Pre-answer is generated *once per query*, conditioned *only on the question $Q$* and explicitly *without access to any video content* (Prompt$_2$ in Appendix A.7.2). It serves as a query-level semantic prior describing "what the answer should look like". The same Pre-answer is then shared across all $K$ candidates of that query.

$$\text{Pre-answer} = \text{LLM}(Q; \text{Prompt}_2), (\text{shared across } c_1, \ldots, c_K)$$

- **(b) Subtitle[†] (per-candidate).** For each candidate $c_k$, the LLM summarizes *the subtitles falling inside that candidate's time span* (Prompt$_1$ in Appendix A.7.2), yielding a candidate-specific $C^k_{\text{vis\_Subtitle}}$ for $k = 1, \ldots, K$.

$$C^k_{\text{vis\_Subtitle}} = \text{LLM}(\text{subtitles}(c_k); \text{Prompt}_1), k = 1, \ldots, K$$

**Stage 3: Iterative Temporal Reasoning and Validation** The $k$ candidate segments, along with their corresponding Subtitle[†] and pre-answers, are sequentially fed into the CACR model in the order provided by VBCS. For each candidate, CACR performs cross-modal reasoning based on the provided information and outputs a timestamp $[t^*_s, t^*_e]$. The validity of the timestamp is verified by $t_s, t_e \in \mathbb{R}^+$. If the timestamp is valid, it is immediately returned as the final prediction. If the output is $[0, 0]$, indicating rejection, the model proceeds to evaluate the next candidate in the sequence. The process repeats until it finds a valid timestamp.

# 4. Experiments

## 4.1. Experimental Settings

**Datasets.** We conduct a comprehensive evaluation on six challenging instructional video datasets: CMIVQA (Li et al., 2023; 2025), MedVidQA (Gupta et al., 2023), VehicleVQA (Luo et al., 2019), TutorialVQA (Colas et al., 2019), COIN (Tang et al., 2021b), and CrossTask (Zhukov et al., 2019). These datasets span a diverse range of domains. Among them, CMIVQA, MedVidQA, VehicleVQA, and TutorialVQA focus on specialized fields, including medical, automotive, and software editing. In contrast, COIN and CrossTask broaden the evaluation to everyday activities and

common tasks. Specifically, COIN covers 12 categories such as vehicle repair, sports, and crafts, while CrossTask includes scenarios like cooking, home repair, and vehicle maintenance.

These datasets encompass a wide range of scenarios from professional domains to everyday applications, enabling a thorough examination of the model's robustness and generalization capabilities (see Appendix A.5). In terms of temporal reasoning complexity, VehicleVQA(Luo et al., 2019) exhibits the smallest Dur./Span value (5.93), reflecting that its tasks require relatively coarse-grained temporal localization. In contrast, TutorialVQA(Colas et al., 2019) presents a significantly higher Dur./Span ratio of 48.9, indicating that the model must, on average, pinpoint a 1-second answer segment within a 49-second video. This demands highly efficient long-sequence search and fine-grained temporal localization capabilities. The detailed statistics of all datasets are summarized in Table 1, with extended statistics in Appendix A.4.

*Table 1.* Statistics of the instructional video datasets used in our experiments. Duration values are reported in seconds.

| Dataset | Domain | Videos | QA Pairs | Duration Range (s) | Avg. Dur. (s) | Avg. Span (s) | Dur./Span |
|---|---|---|---|---|---|---|---|
| VehicleVQA(Luo et al., 2019) | Automotive | 107 | 8,632 | 38.87–309.06 | 125.27 | 21.12 | 5.93 |
| TutorialVQA(Colas et al., 2019) | Software Editing | 76 | 6,195 | 83.40–588.35 | 284.12 | 5.81 | 48.90 |
| MedVidQA(Gupta et al., 2023) | Medical | 899 | 3,010 | 32.99–1,596.68 | 415.99 | 61.97 | 6.71 |
| CMIVQA(Li et al., 2025) | Medical | 1,628 | 7,880 | 28.00–1,933.66 | 311.98 | 33.36 | 9.35 |
| COIN(Tang et al., 2021b) | Daily Life | 11,827 | 46,354 | 4.41–1236.25 | 142.34 | 14.91 | 9.55 |
| CrossTask(Zhukov et al., 2019) | Daily Tasks | 4,700 | 32,426 | 22.00–1118.00 | 296.60 | 9.01 | 32.91 |

**Implementation Details.** We adopt Qwen2.5-VL-7B-Instruct as the base model for its robust visual-language understanding capabilities. For video frame sampling, we employ an adaptive 2 FPS strategy that preserves critical temporal information while managing computational load. Input frames are resized to 448×448 pixels. Training is conducted with batch size 1 for 1-3 epochs on NVIDIA H20 GPUs. Key hyperparameters include: reward balance coefficient $\alpha = 0.8$ and KL penalty coefficient $\beta = 0.1$. Our framework is not sensitive to the choice of LLM. As long as a modern LLM with strong reasoning capabilities is used (such as the four tested in A.7.1), the method delivers similarly strong and stable performance across tasks.

**Evaluation Metrics.** Following standard practice in temporal localization evaluation (Weng & Li, 2023; Li et al., 2024b; Kusa et al., 2022), we adopt two standard metrics: **R@1, IoU=$\mu$** (Recall@1 at IoU threshold $\mu \in \{0.3, 0.5, 0.7\}$) and mIoU (mean Intersection over Union). These metrics quantitatively assess the temporal alignment between predicted segments and ground-truth annotations, ensuring fair comparison with existing methods.

## 4.2. Main Results

As shown in Table 2, CACR achieves state-of-the-art or highly competitive performance across most datasets and

IoU thresholds. The results reveal distinct methodological characteristics under varying video duration ratios.

Based on a comprehensive evaluation across six instructional video datasets, the CACR method demonstrates consistently superior performance in tasks spanning diverse domains and complexity levels, as presented in Table 2. Existing methods exhibit clear performance stratification when faced with different types of challenges. LVLMs that heavily rely on dense frame-level visual features—such as Qwen2.5-VL-7B-Instruct (Bai et al., 2025), TimeZero (Wang et al., 2025b), Time-R1 (Wang et al., 2025c), and VTG-R1 (Dong et al., 2025) show limited capability in temporal modeling for long videos. This weakness is particularly pronounced on TutorialVQA (Colas et al., 2019), which has an extremely high Duration-to-Span ratio (48.9). On this dataset, their mIoU scores are significantly lower than that of CACR, indicating inefficient long-sequence search. On the other hand, Vision-Language Pre-training (VLP) based methods like MutualSL (Weng & Li, 2023), while achieving strong performance on most datasets through effective semantic alignment, are sensitive to textual redundancy. Their performance drops notably on CMIVQA (Li et al., 2025)—a dataset containing substantial subtitle noise—where MutualSL (Weng & Li, 2023) achieves an mIoU of only 33.42, revealing a vulnerability to noisy text. Pure LLM-based approaches such as GPT-3.5 (OpenAI, 2023) and GPT-3.5 (CoT) (Wei et al., 2022), which rely solely on subtitle information for temporal localization, struggle to understand the correlation between question semantics and subtitle content. This is evident from their very low mIoU scores on TutorialVQA (Colas et al., 2019) (1.18 and 1.13, respectively).

In contrast, CACR achieves an effective balance between visual cue utilization and semantic alignment through its candidate-aware reasoning mechanism. On TutorialVQA(Colas et al., 2019) with its extreme length ratio, CACR attains an mIoU of 43.65, substantially outperforming all compared methods. Simultaneously, on CMIVQA(Li et al., 2025) with significant textual redundancy, it achieves an mIoU of 45.37—an improvement of over 11 percentage points compared to MutualSL (Weng & Li, 2023). Importantly, the performance gap between these two challenging datasets is only 1.72 pp for CACR, highlighting its strong robustness against diverse difficulties. Furthermore, CACR maintains leading performance on the remaining four datasets, notably achieving mIoU scores of 72.11 on VehicleVQA(Luo et al., 2019) and 41.06 on COIN(Tang et al., 2021b), which considerably surpass the best existing methods. The systematic advantage across all six datasets collectively validates the effectiveness and generalization capability of the proposed CACR framework for complex, long-form video temporal grounding tasks.

## 4.3. Ablation Studies and Analyses

To systematically investigate the impact of the Top-K candidate segments and auxiliary semantic information—including video subtitle, question-based pre-answers, and frame-level descriptions—on temporal grounding performance, we conduct a comprehensive ablation study covering both top-$K$ selection (on four datasets) and component combinations (on three datasets). In the following, we separately analyze the selection of the hyper-parameter K and the contribution of each semantic component.

### 4.3.1. SELECTION OF THE TOP-$K$ CANDIDATE NUMBER

To determine the optimal number of candidate segments $K$, we systematically evaluate the IoU between the top-$K$ candidate segments generated by VBCS and the ground-truth answer segments across multiple datasets. The analysis employs two complementary metrics: the independent average IoU of candidates at each rank (Table 4), and the cumulative maximum IoU considering the best segment within the top-$K$ candidates (Table 5).

*Table 4.* Independent IoU analysis: Average IoU (%) between candidate segments at each rank and the ground-truth.

| Dataset | Top1 | Top2 | Top3 | Top4 | Top5 | Top6 | Top7 | Top8 | Top9 |
|---|---|---|---|---|---|---|---|---|---|
| VehicleVQA | 43.82 | 41.01 | 37.04 | 34.53 | 31.53 | 29.21 | 27.45 | 25.98 | 24.73 |
| TutorialVQA(Colas et al., 2019) | 43.45 | 10.20 | 5.47 | 5.30 | 4.98 | 4.52 | 4.34 | 4.15 | 3.87 |
| MedVidQA | 58.32 | 53.04 | 49.14 | 42.72 | 32.87 | 28.15 | 25.43 | 23.52 | 21.96 |
| CMIVQA | 36.33 | 31.09 | 26.33 | 23.59 | 21.21 | 19.47 | 18.12 | 17.01 | 16.08 |

As shown in Table 4, the independent IoU exhibits a clear declining trend as the rank decreases from Top-1 to Top-9 across all datasets. This confirms the validity of VBCS's ranking mechanism, as higher-ranked candidates generally possess better quality.

*Table 5.* Cumulative IoU analysis: Maximum IoU (%) achievable by selecting the best segment from Top-K candidates.

| Dataset | $IoU_1$ | $IoU_2$ | $IoU_3$ | $IoU_4$ | $IoU_5$ | $IoU_6$ | $IoU_7$ | $IoU_8$ | $IoU_9$ |
|---|---|---|---|---|---|---|---|---|---|
| VehicleVQA | 43.82 | 45.91 | 47.23 | 48.15 | 49.07 | 49.82 | 50.41 | 50.89 | 51.32 |
| TutorialVQA | 43.45 | 44.12 | 44.35 | 44.67 | 45.21 | 45.43 | 45.62 | 45.78 | 45.91 |
| MedVidQA | 58.32 | 60.14 | 61.27 | 62.05 | 62.73 | 63.25 | 63.68 | 64.02 | 64.31 |
| CMIVQA | 36.33 | 38.45 | 39.87 | 40.92 | 41.76 | 42.43 | 42.98 | 43.45 | 43.85 |

Table 5 shows that the cumulative maximum IoU ($IoU_K$) increases monotonically with $K$ but with diminishing marginal gains. Specifically, when $K$ increases from 1 to 5, $IoU_K$ sees significant improvement across datasets. However, further increasing $K$ from 5 to 9 yields considerably smaller gains. The diminishing returns beyond $K = 5$ suggest that while additional candidates provide marginal coverage improvements, they also introduce lower-quality segments that may confuse the reasoning model. This empirical finding aligns with our theoretical expectation that a well-designed candidate generator should concentrate high-quality proposals within a manageable set. Based on this

*Table 2.* Comparison with prior methods on six instructional TAGV benchmarks. Metrics are reported as percentage. Missing entries are not reported by the original papers. Bold values indicate the best performance in each column, while underlined values indicate the second-best performance. Superscripts indicate the training setting on the target dataset: [S] supervised (full / closed-set fine-tuning, including RL post-training such as GRPO/PPO on the target benchmark; [ZS] zero-shot (no training on the target benchmark, including LLM/LVLM-only and prompting-based methods).

| Family | Method | MedVidQA | | | | VehicleVQA | | | | CMIVQA | | | | TutorialVQA | | | | COIN | | | | CrossTask | | | |
|---|---|---|---|---|---|---|---|---|---|---|---|---|---|---|---|---|---|---|---|---|---|---|---|---|---|
| | | R@0.3 | R@0.5 | R@0.7 | mIoU | R@0.3 | R@0.5 | R@0.7 | mIoU | R@0.3 | R@0.5 | R@0.7 | mIoU | R@0.3 | R@0.5 | R@0.7 | mIoU | R@0.3 | R@0.5 | R@0.7 | mIoU | R@0.3 | R@0.5 | R@0.7 | mIoU |
| **Random** | Random Mode. | 8.38 | 1.93 | 1.21 | 6.89 | 6.52 | 2.75 | 1.54 | 5.22 | 5.71 | 4.65 | 3.58 | 3.97 | 6.53 | 2.46 | 0.00 | 5.26 | 4.92 | 0.76 | 0.28 | 3.12 | 3.57 | 0.54 | 0.20 | 2.79 |
| **VLP** | VSLBase[S] (Zhang et al., 2021) | 27.66 | 14.19 | 6.99 | 21.01 | 18.95 | 8.64 | 4.28 | 20.11 | — | — | — | — | 10.84 | 9.58 | 0.37 | 8.71 | — | — | — | — | — | — | — | — |
| | VSLNet[S] (Zhang et al., 2020a) | 30.32 | 16.61 | 8.39 | 22.41 | 16.53 | 8.47 | 4.03 | 20.07 | — | — | — | — | 11.03 | 9.93 | 0.66 | 9.58 | 13.11 | 7.99 | 4.36 | 9.06 | 19.04 | 12.76 | 8.32 | 13.22 |
| | TMLGA[S] (Rodriguez-Opazo et al., 2020) | 23.87 | 14.84 | 6.21 | 20.49 | 17.69 | 8.79 | 3.46 | 16.54 | — | — | — | — | 12.61 | 11.37 | 1.26 | 8.65 | — | — | — | — | — | — | — | — |
| | ACRM[S] (Tang et al., 2021a) | 24.83 | 16.55 | 10.96 | 22.89 | 20.77 | 12.10 | 8.27 | 22.28 | — | — | — | — | 12.61 | 11.37 | 1.26 | 11.12 | — | — | — | — | — | — | — | — |
| | MoR[S] (Kusa et al., 2022) | 47.10 | 27.74 | 10.97 | 30.67 | 42.75 | 31.54 | 25.97 | 44.81 | — | — | — | — | 23.45 | 15.01 | 8.55 | 19.48 | 14.64 | 8.81 | 3.64 | 10.66 | 23.66 | 18.83 | 10.96 | 16.92 |
| | VTPSL[S] (Li et al., 2024acb) | 77.42 | 61.94 | 44.52 | 57.81 | 74.15 | 67.15 | 54.59 | 64.51 | 40.55 | 29.11 | 14.54 | 28.98 | 50.07 | 40.01 | 25.79 | 40.20 | 30.13 | 18.67 | 9.39 | 22.09 | 42.94 | 32.32 | **36.60** | **25.48** | 31.69 |
| | MutualSL[S] (Weng & Li, 2023) | **80.65** | 61.94 | 39.99 | _58.32_ | _78.74_ | _69.81_ | _54.59_ | _65.74_ | 46.89 | 30.92 | 17.91 | 33.42 | _60.14_ | _43.59_ | _28.28_ | _43.48_ | _55.37_ | _39.49_ | _21.73_ | 38.36 | _49.59_ | 32.32 | 17.04 | _35.33_ |
| | Ouc AI[S] (Zhang et al., 2024) | — | — | — | — | — | — | — | — | 50.88 | 35.42 | 20.54 | 36.37 | | | | | | | | | | | | |
| | SETAG[S] (Zhou et al., 2023) | — | — | — | — | — | — | — | — | 47.75 | 32.09 | 18.98 | 33.89 | | | | | | | | | | | | |
| **LLM** | GPT-3.5[ZS] (OpenAI, 2023) | 52.90 | 41.29 | 22.58 | 38.69 | 30.67 | 15.00 | 8.33 | 24.93 | **63.33** | _47.00_ | _25.67_ | _43.87_ | 2.33 | 0.00 | 0.00 | 1.18 | 20.56 | 10.28 | 3.74 | 15.40 | 22.85 | 10.15 | 3.97 | 14.48 |
| | SubGPT[ZS] (Xiao et al., 2025) | 76.90 | _63.60_ | _44.80_ | 58.00 | — | — | — | — | — | — | — | — | — | — | — | — | 50.90 | 36.40 | 21.40 | _38.40_ | — | — | — | — |
| **LLM-CoT** | GPT-3.5 (CoT)[ZS] (Wei et al., 2022) | 61.29 | 47.10 | 25.16 | 43.44 | 36.33 | 19.00 | 7.00 | 27.79 | 56.33 | 40.00 | 20.33 | 39.16 | 1.67 | 0.33 | 0.00 | 1.13 | — | — | — | — | — | — | — | — |
| **LVLM** | Qwen2.5-VL-7B-Instruct[ZS] (Bai et al., 2025) | 9.70 | 4.55 | 1.52 | 7.03 | 8.49 | 4.72 | 2.83 | 7.03 | 15.38 | 9.83 | 4.70 | 12.21 | 7.90 | 3.85 | 2.12 | 5.39 | 11.63 | 5.22 | 3.20 | 8.83 | 6.52 | 3.44 | 1.90 | 6.05 |
| | TimeZero[S] (Wang et al., 2025b) | 40.98 | 31.14 | 18.85 | 31.34 | 58.60 | 42.87 | 27.01 | 42.41 | 42.58 | 25.66 | 11.86 | 29.22 | 22.90 | 10.01 | 4.35 | 18.44 | 29.26 | 15.89 | 5.36 | 20.05 | 27.15 | 14.75 | 4.46 | 18.57 |
| | TFVTG[ZS] (Zheng et al., 2024) | 35.42 | 28.37 | 19.92 | 27.60 | 47.98 | 37.80 | 22.10 | 35.45 | 29.94 | 16.01 | 9.88 | 22.44 | 19.34 | 9.16 | 4.58 | 13.58 | 21.15 | 15.98 | 8.97 | 17.95 | 22.16 | 10.97 | 3.45 | 14.14 |
| | Time-R1[S] (Wang et al., 2025c) | 42.25 | 35.50 | 23.52 | 37.38 | 59.06 | 44.27 | 29.36 | 44.28 | 43.45 | 28.36 | 15.09 | 32.06 | 22.15 | 13.98 | 7.97 | 19.95 | 31.82 | 16.97 | 10.21 | 22.25 | 27.39 | 13.85 | 4.82 | 18.97 |
| | VTG-R1[S] (Dong et al., 2025) | 34.78 | 26.59 | 12.41 | 27.59 | 53.24 | 40.80 | 26.56 | 38.99 | 38.95 | 20.35 | 7.22 | 25.93 | 18.28 | 9.89 | 3.79 | 14.38 | 27.09 | 13.97 | 4.70 | 18.85 | 23.64 | 9.27 | 2.42 | 15.03 |
| | Timescope[S] (Liu et al., 2025) | 46.49 | 39.27 | 24.52 | 39.72 | 65.74 | 50.87 | 32.09 | 48.45 | 43.75 | 23.39 | 8.06 | 31.61 | 26.01 | 12.90 | 5.24 | 18.59 | 29.87 | 16.85 | 9.30 | 21.44 | 26.85 | 15.48 | 5.73 | 19.70 |
| | Ask2Loc[S] (Zong et al., 2025) | 62.78 | 33.37 | 23.57 | 43.22 | 67.42 | 52.65 | 35.54 | 55.28 | 42.46 | 25.04 | 17.14 | 31.37 | — | — | — | — | — | — | — | — | — | — | — | — |
| | Ours[S] | _79.91_ | **67.09** | **45.87** | **59.65** | **85.31** | **76.70** | **69.33** | **72.11** | _62.50_ | **48.52** | **33.82** | **45.37** | **60.31** | **43.65** | **28.72** | **43.65** | **59.11** | **43.81** | **24.88** | **41.06** | **55.89** | **41.77** | _22.40_ | **40.19** |

*Table 3.* Comparison of different components on three instructional TAGV benchmarks. Metrics are reported as percentage; mIoU is the mean Intersection-over-Union over the test set (also expressed as a percentage). Bold values indicate the best performance in each column, while underlined values indicate the second-best performance. Subtitle† denotes the LLM-summarized version of the dataset-provided subtitles within each candidate segment

| Method | VBCS | Pre-answer | Subtitle† | Description | MedVidQA | | | | VehicleVQA | | | | CMIVQA | | | |
|---|---|---|---|---|---|---|---|---|---|---|---|---|---|---|---|---|
| | | | | | R@0.3 | R@0.5 | R@0.7 | mIoU | R@0.3 | R@0.5 | R@0.7 | mIoU | R@0.3 | R@0.5 | R@0.7 | mIoU |
| TimeZero (Wang et al., 2025b) | | | | | 40.98 | 31.14 | 18.85 | 31.34 | 58.60 | 42.87 | 27.01 | 42.41 | 42.58 | 25.66 | 11.86 | 29.22 |
| TimeZero + Subtitle† (Wang et al., 2025b) | | | ✓ | | 45.21 | 33.78 | 20.15 | 33.58 | 61.35 | 45.92 | 28.73 | 44.32 | 40.17 | 24.35 | 11.92 | 27.85 |
| TimeZero + Pre-Answer (Wang et al., 2025b) | | ✓ | | | 43.87 | 32.45 | 19.63 | 32.68 | 60.12 | 44.36 | 27.89 | 43.52 | 39.65 | 23.87 | 11.45 | 27.12 |
| TimeZero + Subtitle† + Pre-Answer (Wang et al., 2025b) | | ✓ | ✓ | | 47.35 | 35.92 | 21.78 | 35.42 | 63.78 | 48.15 | 30.45 | 46.25 | 43.11 | 26.97 | 12.75 | 30.45 |
| MutualSL (Weng & Li, 2023) | ✓ | | | | **80.65** | 61.94 | 39.99 | 58.32 | 78.74 | 69.81 | 53.14 | 65.74 | 46.89 | 30.92 | 17.91 | 33.42 |
| CACR-TopK (Top-K) | ✓ | | | | 75.74 | _63.24_ | 45.59 | _59.04_ | 73.74 | 71.07 | 65.09 | 62.88 | 57.51 | 41.59 | 22.07 | 39.01 |
| CACR-Cap (Top-K + Subtitle†) | ✓ | | ✓ | | 78.68 | 61.76 | 42.65 | 57.63 | 78.54 | _76.68_ | _69.10_ | 65.39 | 56.01 | 40.95 | 21.37 | 38.74 |
| CACR-Pre (Top-K + Pre-Answer) | ✓ | ✓ | | | 75.00 | 60.29 | _46.32_ | 57.30 | 79.32 | 72.15 | 64.25 | _70.85_ | 53.36 | 38.25 | 22.95 | 37.08 |
| CACR-DES (Top-K + Description) | ✓ | | | ✓ | 74.67 | 60.34 | 45.80 | 58.52 | 80.67 | 72.37 | 57.86 | 65.20 | 56.23 | 39.79 | 20.58 | 38.93 |
| CACR-Group1 (Top-K + Subtitle† + Description) | ✓ | | ✓ | ✓ | 74.11 | 60.73 | **46.63** | 58.65 | 82.00 | 72.37 | 59.12 | 66.19 | 59.24 | 40.64 | 23.06 | 40.52 |
| CACR-Group2 (Top-K + Pre-Answer + Description) | ✓ | ✓ | | ✓ | 74.94 | 60.79 | 44.85 | 57.65 | 83.80 | 75.14 | 61.83 | 68.2 | 59.74 | 39.92 | 21.79 | 40.22 |
| CACR-Group3 (Top-K + Pre-Answer + Description + Subtitle†) | ✓ | ✓ | ✓ | ✓ | 74.33 | 58.79 | 41.52 | 55.63 | _84.35_ | 76.22 | 62.79 | 69.04 | _60.26_ | _42.62_ | _24.74_ | _44.60_ |
| CACR-Group4 (Top-K + Subtitle† + Pre-Answer) | ✓ | ✓ | ✓ | | _79.91_ | **67.09** | 45.87 | **59.65** | **85.31** | **76.70** | **69.33** | **72.11** | **62.50** | **48.52** | **33.82** | **45.37** |

analysis, we set $K = 5$ for all subsequent experiments.

## 4.4. Necessity Analysis of Top-K Candidate Segments and Auxiliary Semantic Information

As shown in Table 3, Top-K candidates form the foundation for high-performance temporal localization, while auxiliary semantic information further significantly enhances reasoning accuracy and robustness. The importance of Top-K candidates is clearly demonstrated by comparing CACR-TopK with the TimeZero variants. Across all three datasets, CACR-TopK consistently and substantially outperforms every TimeZero-based method. This marked improvement confirms the critical efficacy of the VBCS candidate generation mechanism in compressing the search space and providing a high-quality starting point for localization. To systematically evaluate the role of different auxiliary semantic information, this study introduces three comparative components: visual detail description of video frames (Description), original subtitle information from the video (summary of subtitles, i.e., summarized character dialogue/narration), and pre-answers generated based on the

questions (Pre-Answer). The prompts used are detailed in Appendix A.7.2. By constructing multiple groups of comparative models with different component combinations, the synergistic mechanisms and necessity of each semantic information source are comprehensively validated.

Experimental results on the datasets show that subtitles, as the dialogue or narration within the video, provides direct textual clues for event development; whereas Pre-Answer focuses on causal reasoning concerning "why" and "how," highlighting key information relevant to the question's logic. These two components form functional complementarity at the levels of "direct content presentation" and "causal reasoning guidance," effectively mitigating the comprehension bias or limitations that may arise from a single information source. In contrast, Description, which is descriptive text generated for key frame content, contributes relatively less within this synergistic framework—overly detailed visual descriptions can introduce information redundancy and noise, potentially interfering with the precise localization of temporal structures.

In summary, through the systematic comparison of multiple

auxiliary semantic information sources, this section confirms that Top-K candidate segment screening serves as a necessary foundation for effectively leveraging semantic priors, while the complementary integration of original subtitles and Pre-Answer achieves optimal synergy at the levels of direct content clues and high-level causal reasoning. This thereby provides robust and accurate semantic guidance for the temporal localization task in video question answering.

## 5. Conclusion

In this work, we introduce CACR, an innovative framework designed to address core challenges in temporal answer grounding in instructional video (TAGV) using a candidate-aware causal reasoning approach. Our contributions are threefold: first, we propose the VBCS algorithm to efficiently generate candidate segments from long videos, effectively mitigating the extreme ratio between video duration and answer segment length; second, we develop a GRPO-optimized reasoning module enhanced by a differentiated prompting strategy and a composite reward function that integrates temporal alignment and rejection signals, substantially improving reasoning robustness; third, extensive evaluations on six challenging datasets show that CACR achieves state-of-the-art mIoU on all six benchmarks, especially in handling extreme duration ratios (TutorialVQA(Colas et al., 2019)) and textual redundancy (CMIVQA(Li et al., 2025)). Ablation studies further confirm the necessity of each component and their synergistic effects. Future work will extend this framework to multi-step temporal reasoning and cross-modal alignment in more complex video understanding scenarios.

## Acknowledgements

This work was supported by the Shenzhen Medical Research Fund (No. D2404001), and in part by the National Key Laboratory of the CAS on Medical Imaging Science and Technology System, the Key Research and Development Program of Guangdong Province (No. 2025B1111020001), the Shenzhen STIB programs (No. CJGJZD20230724093303007 and KJZD20240903101259001), and the Xisike Clinical Oncology Research Foundation (Y-2024AZ(NSCLC)MS-0156).

## Impact Statement

This study addresses video understanding and temporal localization tasks using publicly available instructional video datasets (VehicleVQA(Luo et al., 2019), TutorialVQA(Colas et al., 2019), MedVidQA(Gupta et al., 2023), CMIVQA(Li et al., 2025),COIN(Tang et al., 2021b),CrossTask(Zhukov et al., 2019)). All datasets represent benchmark resources from the research community that have undergone appropriate de-identification procedures and contain no personally sensitive information. The video data originate from publicly accessible sources, with collection processes adhering to the ethical guidelines of original providers. This research involves no new human subject data collection or animal experimentation. The video content primarily features instructional demonstrations (e.g., automotive repair, software operation, medical procedures) without ethically controversial material.

Our proposed method enhances video content understanding accuracy for potential applications in educational assistance and knowledge retrieval. We mitigate model biases through multi-dataset validation and incorporate temporal logic considerations to prevent misleading outcomes. This research remains exploratory and is not recommended for clinical diagnosis or high-risk decision-making scenarios. Overall, this study presents no significant ethical risks and complies with academic research standards.

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

# A. *Appendix*

### A.1. Case Study

To demonstrate the operational efficacy of our inference pipeline, we present a concrete case study. The input consists of a video $V$ and a question $Q$: "How to check if your crutches fit properly to you." The ground-truth solution timestamp is $[21, 88]$.

**Stage 1: Candidate Proposal Generation.** The VBCS processes the input video and proposes $k = 5$ candidate temporal segments. These proposals are generated and sorted in descending order based on the model's predicted confidence scores, forming a prioritized list for subsequent evaluation. The candidates are as follows:

- Candidate 1: $[173.37, 317.59]$

- Candidate 2: $[20.24, 179.29]$

- Candidate 3: $[24.47, 187.32]$

- Candidate 4: $[39.24, 173.37]$

- Candidate 5: $[16.46, 210.16]$

**Stage 2: Subtitle[†] and Pre-Answer Generation.** For each candidate segment, the corresponding video clip and the question $Q$ are fed into the LLM, which generates a descriptive subtitle and a preliminary answer (pre-answer). As the question is identical for all candidates, the pre-answer remains consistent: "To check if crutches fit properly: Stand with your arms at your sides. The top of the crutch should reach the fold of your armpit, with about two fingers' width between. Your elbows should be slightly bent. When you hold the handgrips, they should be at hip level. Adjust the length if needed to ensure comfortable, pain-free use."

The generated subtitle summaries for each candidate are:

- **Candidate 1 ([173.37, 317.59])**: "Instructions on using crutches: Fit crutches correctly, stand tall, bring one crutch to the other side. Move crutches forward first, then swing the injured leg and step with the good leg. For stairs, adjust crutches and use the railing. Follow doctor's weight-bearing advice, and don't lean on crutches with armpits, stand tall, and move crutches before stepping."

- **Candidate 2 ([20.24, 179.29])**: "When using crutches for the first time, ensure they fit properly. Check height with a tool, align wrist part with wrist, have a slight elbow bend and 2 - finger-width gap under armpits. Demonstrate getting up from chair/bed (using injured leg forward, pushing with hands) and sitting down (backing up, repositioning crutches)"

- **Candidate 3 ([24.47, 187.32])**: "The video goes through crutch basics. Check the height-setting tool, ensure correct height with a gap between armpit and crutch top. Don't lean on the pads. Demonstrate transferring from chair/bed to standing and vice versa, emphasizing proper technique to avoid nerve damage and maintain balance."

- **Candidate 4 ([39.239, 173.37])**: "The text details how to adjust crutches to the right height. Keep a two - finger - width gap between armpit and crutch top, with elbows slightly bent when gripping. Demonstrate standing up from a chair (using the affected leg forward with crutches on that side) and sitting down, ensuring proper balance and controlled movements."

- **Candidate 5 ([16.46, 210.159])**: "This video gives basic tips on using crutches correctly. Check height with a tool, ensure proper fit at wrist and elbow, keep a gap under armpit. Demonstrate getting up from chair/bed, sitting down, walking with crutches, and going up/down stairs, emphasizing correct weight-bearing and movement techniques."

**Stage 3: Iterative Temporal Reasoning and Validation.** The CACR model processes the candidates sequentially.

1. **For Candidate 1 ([173.37, 317.59])**, the model receives the corresponding subtitle and pre-answer. The reasoning output is: `"ans": "<think> The given video content does not contain the`

specific information about how to check if crutches fit properly to someone. The video provides instructions on using crutches, going up and down stairs, and general advice but does not address the fitting aspect.</think>\n<answer>0 to 0.00</answer>". This yields an invalid timestamp $[0.0]$, prompting the model to reject this candidate and proceed.

2. **For Candidate 2 ([20.24, 179.29])**, the model performs cross-modal reasoning based on its subtitle and the shared pre-answer. The output is: "ans": "<think> The segment starts by showing the woman adjusting her crutches to ensure they fit properly. This includes checking the height, aligning the wrist part with the wrist, having a slight elbow bend, and ensuring there is a 2-finger-width gap under the armpits. These actions directly address how to check if the crutches fit properly.\n\n</think>\n<answer>0 to 65.00</answer>". This timestamp $[0, 65.00]$ is valid within the candidate's local timeframe. It is then mapped back to the original video's timeline: $[0 + 20.24, 65.00 + 20.24] = [20.24, 85.24]$. This valid timestamp $[20.24, 85.24]$ is immediately returned as the final prediction, and the inference process terminates.

This case illustrates the pipeline's ability to iteratively validate proposals. The first, irrelevant candidate is efficiently rejected. The second candidate is accepted as it contains the pertinent visual evidence aligned with the query. The final predicted interval $[20.24, 85.24]$ shows high overlap with the ground-truth $[21, 88]$, raising the mIoU from 42.13% ((based on Candidate 2's raw bounds $[20.24, 179.29]$, IoU=42.13%) to 94.8%, which demonstrates the critical role of the CACR's fine-grained reasoning in enhancing temporal localization accuracy.

### A.2. MutualSL (Weng & Li, 2023) Method Description

Due to space limitations in the main text, we provide here a comprehensive explanation of MutualSL's principles and key components.

The VBCS framework, instantiated with MutualSL (Weng & Li, 2023), operates through four sequential stages to generate top-$K$ candidate segments for video moment retrieval. This approach effectively addresses the length disparity challenge by narrowing the processing scope for subsequent modules while maintaining high recall of ground-truth segments.

STAGE 1: CROSS-MODAL FEATURE EXTRACTION AND FUSION

The process begins with extracting and fusing multi-modal features to establish semantic connections between visual content and textual information:

$$\mathbf{V} = \text{I3D}(V) \in T \tag{3}$$

$$\mathbf{T} = \text{PLM}([Q, T_1, \ldots, T_r]) \in^{n \times d} \tag{4}$$

Visual features are extracted using a pre-trained I3D model, where $k$ represents the number of video frames and $d = 1024$ denotes the feature dimension, encoding spatiotemporal dynamics of the video. Textual features are derived from concatenated queries and subtitles through a pre-trained language model, with $n$ indicating the number of text tokens, ensuring dimensional consistency with visual features for subsequent fusion.

Cross-modal fusion employs context-query attention mechanisms to enhance semantic interaction:

$$\mathcal{D} = \mathcal{G}_r \cdot \mathbf{T} \tag{5}$$

$$\mathcal{F} = \mathcal{G}_c \cdot \mathcal{G}_r^T \cdot \mathbf{V} \tag{6}$$

$$V'' = \text{Conv1d}\big(\text{Concat}[\text{Attention}(V', \mathbf{T}); \mathbf{T}]\big) \tag{7}$$

The visual pathway fusion computes context-to-query attention $\mathcal{D}$ using weight matrix $\mathcal{G}_r$ that measures video frames' relevance to textual queries, and query-to-context attention $\mathcal{F}$ using weight matrix $\mathcal{G}_c$ that captures textual focus on visual

frames. The enhanced visual features $V''$ are obtained through 1D convolutional processing of concatenated attention features.

$$\bar{T} = \{\text{AvgPool}(V'') + T'_j\}_{j=1}^n \in^{n \times d} \tag{8}$$

The textual pathway fusion integrates global visual context into each text token through broadcasting, enabling text representations to perceive visual scenes and reducing the modality gap.

STAGE 2: DUAL-PREDICTOR PROBABILITY ESTIMATION

The framework employs separate predictors for visual and textual modalities to generate temporal segment predictions:

$$V_s^{\text{Logits}} = \text{FFN}_{\text{Start}}^{\text{Visual}}\left(\text{LSTM}_{\text{Start}}(V'')\right) \tag{9}$$

$$V_e^{\text{Logits}} = \text{FFN}_{\text{End}}^{\text{Visual}}\left(\text{LSTM}_{\text{End}}(V'')\right) \tag{10}$$

The visual predictor utilizes LSTMs to capture temporal dependencies in fused visual features $V''$, with feed-forward networks generating frame-level probability distributions for start and end boundaries.

$$T_s^{\text{Logits}} = \text{FFN}_{\text{Start}}^{\text{Textual}}(\bar{T}) \tag{11}$$

$$T_e^{\text{Logits}} = \text{FFN}_{\text{End}}^{\text{Textual}}(\bar{T}) \tag{12}$$

The textual predictor employs a QANet-like structure on visually-enhanced text features $\bar{T}$ to produce token-level probability distributions for subtitle span boundaries.

STAGE 3: CROSS-MODAL ALIGNMENT AND MUTUAL KNOWLEDGE TRANSFER

This stage addresses the dimensional mismatch between visual and textual predictions through timeline mapping and mutual learning:

$$\tilde{T}_s = \arg\min_i |V_s - \mathbb{Q}(T_i).\text{center}| \tag{13}$$

$$\tilde{T}_e = \arg\min_i |V_e - \mathbb{Q}(T_i).\text{center}| \tag{14}$$

Timeline mapping $\mathbb{Q}$, implemented as a subtitle-video time correspondence lookup table, enables cross-modal alignment by converting visual frame times to subtitle spans through minimum distance matching. For instance, if subtitle $T_3$ corresponds to video time interval [1:32, 1:50], then $\mathbb{Q}(T_3) = [1 : 32, 1 : 50]$.

$$\tilde{V}_s = \arg\min_i |T_s - \mathbb{Q}(V_i).\text{center}| \tag{15}$$

$$\tilde{V}_e = \arg\min_i |T_e - \mathbb{Q}(V_i).\text{center}| \tag{16}$$

Conversely, textual subtitle spans are converted to visual frame times using the same mapping principle, enabling seamless knowledge exchange between modalities.

The training objective combines base prediction loss and mutual transfer loss:

$$\text{Loss}_{\text{Visual}} = \text{CE}(V_s^{\text{Logits}}, V_s^{gt}) + \text{CE}(V_e^{\text{Logits}}, V_e^{gt}) \tag{17}$$

$$\text{Loss}_{\text{Textual}} = \text{CE}(T_s^{\text{Logits}}, T_s^{gt}) + \text{CE}(T_e^{\text{Logits}}, T_e^{gt}) \tag{18}$$

The base prediction loss ensures individual predictor accuracy by minimizing cross-entropy between predictions and ground-truth labels, where text ground-truth is derived from video labels via timeline mapping.

$$\text{Loss}_{\text{Visual}}^{\text{Mutual}} = \lambda_v \times [\text{CE}(V_s^{\text{Logits}}, (\tilde{V}_s)) + \text{CE}(V_e^{\text{Logits}}, (\tilde{V}_e))] \tag{19}$$

$$\text{Loss}_{\text{Textual}}^{\text{Mutual}} = \lambda_t \times [\text{CE}(T_s^{\text{Logits}}, (\tilde{T}_s)) + \text{CE}(T_e^{\text{Logits}}, (\tilde{T}_e))] \tag{20}$$

The mutual transfer loss facilitates cross-modal knowledge transfer through dynamic weighting and unilateral gradient flow. Here, $\lambda_v = \text{IoU}([\tilde{V}_s, \tilde{V}_e], [V_s^{gt}, V_e^{gt}])$ and $\lambda_t = \text{IoU}([\tilde{T}_s, \tilde{T}_e], [T_s^{gt}, T_e^{gt}])$ are IoU-based weights that prioritize knowledge from more accurate predictions, while  denotes stop-gradient operation that prevents backpropagation to the source predictor.

STAGE 4: TEMPORALLY-EXTENDED CANDIDATE GENERATION AND SELECTION

$$\text{Candidates} = \big\{(\max(0, v_s - \Delta t), \min(v_e + \Delta t, T)) \mid v_s \in \text{Top-}K(V_s), v_e \in \text{Top-}K(V_e), v_s < v_e\big\} \tag{21}$$

During inference, top-$K$ candidate segments are generated by pairing the highest-ranked start and end positions from visual predictions, with temporal extension $\Delta t$ applied to expand segment coverage. The final candidate set $C_{\text{vis}}$ is formed by selecting the first $K$ qualified candidates.

This comprehensive MutualSL methodology enables effective video moment retrieval through synergistic cross-modal collaboration, where timeline mapping and combined losses ensure aligned and mutually refined predictions.

## A.3. Core Mechanism Comparison

CANDIDATE GENERATION: SERIAL EVENT PARSING VS. HYPOTHESIS GENERATION WITH REJECTION

- **TFVTG (Zheng et al., 2024) (for TSGV):** Uses an LLM to parse the query $Q$ into an ordered sequence of sub-events $\{c_1, c_2, \ldots, c_m\}$, then independently matches each $c_i$ to a video segment. This serialized pipeline risks error propagation: if the LLM makes a parsing error (generating an incorrect $c_i$), the error propagates along the chain $\text{Error}(c_i) \rightarrow \text{Error}(S_i) \rightarrow \text{Error}(P^{\text{final}})$ and cannot be corrected in later stages. This paradigm is difficult to adapt to TAGV, as "how-to" questions cannot be simply decomposed into surface-level action sub-events.

- **CACR (for TAGV):** Its candidate generation mechanism is designed for QA-oriented reasoning, aiming to generate a set of potential evidential hypotheses for complex questions. A key innovation is the introduction of a rejection mechanism embedded within a reinforcement learning framework. A specialized reward function optimizes the model to actively veto low-quality candidates, thereby cutting off error propagation chains and significantly enhancing system robustness.

### A.3.1. VIDEO SAMPLING & REASONING PARADIGM: GLOBAL DENSE PROCESSING VS. SPARSE ITERATIVE VERIFICATION

- **VTG-R1 (Dong et al., 2025) / TimeZero(Wang et al., 2025b) / Time-R1(Wang et al., 2025c) / Qwen2.5-VL-7B-Instruct(Bai et al., 2025) (Uniform Sampling Paradigm):** These methods sample the entire long video at a fixed rate $f_{\text{sample}}$ into a uniform frame sequence $F_{\text{uniform}}$, which is fed entirely into the LVLM for direct timestamp regression:

$$\mathcal{I}^* = \text{LVLM}(F_{\text{uniform}}, Q)$$

*Drawbacks:* The number of visual tokens grows linearly with video duration, easily hitting the model's context length limit. The process is computationally intensive, and attention is dispersed over many irrelevant frames.

- **CACR (Candidate-Aware Sparse Iterative Paradigm):**

  1. A lightweight VBCS module filters a small set of candidate segments $\mathcal{C} = \{c_k = (t_s^k, t_e^k)\}$ from the full video, transforming the long-video search into "causal decision-making within a finite candidate set."

  2. For each candidate $c_k$, a denser frame sequence $F_{\text{candidate}}^k$ is extracted using a higher sampling rate $f_{\text{candidate}}$. After

assembling multi-source information, it is iteratively fed to the LVLM for verification:

$$\text{Input}^k_{\text{LVLM}} = \text{Assemble}\big(F^k_{\text{candidate}}, C^k_{\text{vis\_Subtitle}}, \text{Pre-answer}, Q\big)$$
$$o_k = \text{LVLM}\big(\text{Input}^k_{\text{LVLM}}\big)$$

3. **Decision Rule (first-valid with explicit fallback):** Given the candidate set $\mathcal{C} = \{c_1, c_2, \ldots, c_K\}$ ordered by VBCS confidence, the LVLM is invoked sequentially on $c_1, c_2, \ldots, c_K$. For each output $o_k$:

   - if $o_k = [t^*_s, t^*_e]$ with $t^*_s, t^*_e \in \mathbb{R}^+$ (i.e., a valid temporal interval), $o_k$ is *immediately* returned as the final prediction and the iteration terminates;
   - if $o_k = [0.0, 0.0]$ (the rejection token), candidate $c_k$ is skipped and the model proceeds to $c_{k+1}$.

   This is a strict *first-valid* rule: no comparison or re-ranking is performed across candidates, and the predictions of skipped (rejected) candidates are discarded. If *all* $K$ candidates are rejected (an empirically rare event; see Appendix A.1), the system falls back to returning the top-1 VBCS candidate interval $[t^1_s, t^1_e]$ as the prediction, ensuring the pipeline always produces a non-empty output. This rule is fully consistent with the inference pipeline described in Section 3.3 and with the case study in Appendix A.1, where Candidate 1 is rejected and Candidate 2 is accepted and immediately returned without examining Candidate 3.

*Advantage:* This approach combines global sparse screening with local dense analysis, avoiding the computational burden of processing the entire video while preserving visual details in key regions, enabling an efficient "hypothesis-verification" loop.

### A.3.2. MULTI-SOURCE INFORMATION FUSION: ENRICHING THE REASONING CONTEXT

CACR injects dual semantic aids for each candidate segment during inference, providing the LVLM with a rich semantic context far beyond raw frames:

- **Subtitle Summarization:** $C^k_{\text{vis\_Subtitle}} = \text{LLM}(\text{Prompt}_1)$, which summarizes the subtitles within the candidate segment.

- **Pre-answer Generation:** $\text{Pre-answer} = \text{LLM}(\text{Prompt}_2)$, which generates a hypothetical answer based on the question to anchor the reasoning direction.

These two components are complementary, jointly enhancing the model's understanding of abstract questions and localization accuracy.

### A.4. Performance of CACR Across Different Video Duration Intervals

Based on the experimental results presented in Table 6, we draw the following conclusions:

- **Stable cross-duration performance:** The proposed CACR method maintains relatively consistent mean Intersection over Union (mIoU) across various video duration intervals. No significant performance degradation is observed as video length increases, indicating its robustness to duration variation.

- **Strong competitiveness in long-video scenarios:** Particularly for videos exceeding 1000 seconds, CACR achieves an mIoU of 43.60% on CMIVQA (overall 45.37%) and 64.20% on MedVidQA(Gupta et al., 2023)(overall 59.65%). These results demonstrate the method's effectiveness and robustness in processing long video content.

*Table 6.* Performance (mIoU) of CACR across different video duration intervals on four datasets.

| Duration (s) | CMIVQA | | MedVidQA | | VehicleVQA | | TutorialVQA | |
|---|---|---|---|---|---|---|---|---|
| | Sample | mIoU | Sample | mIoU | Sample | mIoU | Sample | mIoU |
| 0–50 | 6 | 41.50% | 1 | 52.00% | 6 | 72.00% | – | – |
| 50–100 | 39 | 42.70% | 7 | 56.80% | 157 | 74.00% | – | – |
| 100–150 | 64 | 45.80% | 15 | 64.40% | 324 | 71.00% | 3 | 38.00% |
| 150–200 | 48 | 37.40% | 2 | 67.10% | 73 | 69.00% | 52 | 38.10% |
| 200–250 | 58 | 47.40% | 3 | 59.30% | – | – | 140 | 37.40% |
| 250–300 | 81 | 47.50% | 8 | 54.60% | 18 | 73.00% | 236 | 43.20% |
| 300–350 | 44 | 52.70% | 3 | 83.10% | 13 | 70.00% | 100 | 46.30% |
| 350–400 | 60 | 42.30% | 8 | 57.30% | – | – | 14 | 39.50% |
| 400–450 | 46 | 47.30% | – | – | – | – | 47 | 52.20% |
| 450–500 | 22 | 46.50% | 11 | 57.70% | – | – | – | – |
| 500–600 | 45 | 44.70% | 18 | 54.40% | – | – | 28 | 65.50% |
| 600–700 | 12 | 45.30% | 9 | 58.70% | – | – | – | – |
| 700–800 | 16 | 43.70% | 14 | 59.40% | – | – | – | – |
| 800–900 | 8 | 45.20% | 13 | 57.70% | – | – | – | – |
| 1000+ | 2 | 43.60% | 9 | 64.20% | – | – | – | – |
| **Overall** | **551** | **45.37%** | **121** | **59.65%** | **591** | **72.11%** | **620** | **43.65%** |

In summary, the above experiments provide strong empirical evidence supporting the potential of CACR in addressing challenges associated with long videos.

## A.5. Experimental Evidence of Robust Performance on COIN and CrossTask

To evaluate the robustness of the proposed CACR framework, we assess its performance across diverse domain scenarios within two representative benchmarks: COIN and CrossTask. The results demonstrate consistent performance with minimal variance, indicating strong generalization capability.

*Table 7.* Robust Performance (mIoU) of CACR on the COIN Dataset Across Different Domains

| Domain | Sample Number | mIoU (%) |
|---|---|---|
| Electrical Appliance | 482 | 41.73 |
| Gadgets | 396 | 39.64 |
| Vehicle | 291 | 41.80 |
| Dish | 242 | 41.28 |
| Furniture and Decoration | 194 | 41.14 |
| Nursing and Care | 184 | 41.10 |
| Leisure and Performance | 183 | 41.09 |
| Pets and Fruit | 174 | 41.05 |
| Science and Craft | 164 | 41.01 |
| Drink and Snack | 187 | 40.99 |
| Housework | 161 | 40.95 |
| Sport | 47 | 40.75 |
| **Total Number** | **2705** | **41.06** |

The experimental results demonstrate CACR's robust performance across diverse domains. On the COIN dataset spanning 12 domains, the maximum variance in mIoU is within 5%. Notably, no domain exhibits performance collapse; even the most challenging 'Sport' domain achieves an mIoU of 40.75%, only 0.31 pp below the cross-domain average of 41.06%; the overall spread across all 12 domains is just 2.16 pp.

The stable and balanced performance observed across internally varied datasets indicates that CACR does not overfit to specific domain characteristics. Its feature extraction and task adaptation mechanisms generalize effectively to both dynamic

*Table 8.* Robust Performance (mIoU) of CACR on the CrossTask Dataset Across Different Domains

| Domain | Sample Number | mIoU (%) |
|---|---|---|
| Cooking | 121 | 38.27 |
| Car Maintenance | 97 | 41.39 |
| Crafting & Home Repairs | 53 | 42.30 |
| **Total Number** | **271** | **40.19** |

action-intensive scenarios (e.g., "Sport" in COIN) and static, object-centric scenarios (e.g., "Home Repair" in CrossTask). These results provide preliminary yet compelling evidence supporting the model's potential for broader application across diverse real-world video understanding tasks. The framework's consistent performance across domains suggests that its causal reasoning approach captures fundamental temporal grounding principles rather than domain-specific artifacts.

### A.6. Statement on the Reasoning of CACR

The term "Causal Reasoning" emphasized in the paper's title does not refer to traditional causal inference methods, but rather to a reasoning mechanism that decomposes the prediction process, introduces semantic mediators, and employs counterfactual validation. This approach shifts the model away from black-box, surface-level association learning based on simple input–output patterns toward deeper reasoning with explicit logical chains, thereby enhancing the interpretability and robustness of the predictions. The following analysis elaborates on this interpretation from both mathematical principles and implementation mechanisms.

#### A.6.1. DECOMPOSITION OF THE REASONING PROCESS

Traditional methods typically model the conditional probability $P([t_s, t_e] \mid V, Q)$ directly, forming a typical black-box prediction problem. This often leads the model to capture superficial statistical associations between the video $V$ and the question $Q$, while overlooking their intrinsic logical connections.

The CACR framework introduces candidate segments $c_k$ and semantic mediators $\text{Semantic} = \{C^k_{\text{vis\_Subtitle}}, \text{Pre-answer}\}$ to decompose the end-to-end prediction into a multi-step conditional reasoning process:

$$P([t_s*, t_e*]|c_k, Q, \text{Semantic}_k) = \sum_{c_k \in C_{\text{vis}}} \underbrace{P([t_s^*, t_e^*] \mid c_k, Q, \text{Semantic})}_{\text{GRPO Causal Validation}} \cdot \underbrace{P(c_k \mid V, Q)}_{\text{VBCS Candidate Generation}} \tag{22}$$

This decomposition constructs a reasoning chain of "Question → Candidate Generation → Context Enrichment → Hypothesis Validation → Answer Decision/Rejection," forcing the model to establish explicit logical connections at each step rather than learning a direct input–output mapping.

#### A.6.2. ROLES OF INDIVIDUAL COMPONENTS

**a. VBCS Module: Candidate Generation with Cross-Modal Alignment**  The Visual-Language Pre-training based Candidate Selection (VBCS) module generates a set of candidate segments $C_{\text{vis}}$ through dual visual–textual predictors, mathematically modeling the probability distribution $P(c_k \mid V, Q)$. Its training objective is a supervised joint loss function:

$$\mathcal{L}_{\text{total}} = \mathcal{L}_{\text{Visual}} + \mathcal{L}_{\text{Textual}} + \mathcal{L}_{\text{Visual}}^{\text{Mutual}} + \mathcal{L}_{\text{Textual}}^{\text{Mutual}} \tag{23}$$

where:

- *Base localization losses* $\mathcal{L}_{\text{Visual}}$ and $\mathcal{L}_{\text{Textual}}$ ensure single-modal boundary accuracy:

$$\mathcal{L}_{\text{Visual}} = \text{CE}(V_s^{\text{Logits}}, V_s^{\text{gt}}) + \text{CE}(V_e^{\text{Logits}}, V_e^{\text{gt}}) \tag{24}$$

$$\mathcal{L}_{\text{Textual}} = \text{CE}(T_s^{\text{Logits}}, T_s^{\text{gt}}) + \text{CE}(T_e^{\text{Logits}}, T_e^{\text{gt}}) \tag{25}$$

- *Cross-modal alignment losses* $\mathcal{L}_{\text{Visual}}^{\text{Mutual}}$ and $\mathcal{L}_{\text{Textual}}^{\text{Mutual}}$ achieve reliable cross-modal knowledge transfer via dynamic weighting and unidirectional gradient stopping. They filter out single-modal noise while preserving cross-modality shared causal features:

$$\mathcal{L}_{\text{Visual}}^{\text{Mutual}} = \lambda_v \times \left[ \text{CE}(V_s^{\text{Logits}}, \text{sg}(\tilde{V}_s)) + \text{CE}(V_e^{\text{Logits}}, \text{sg}(\tilde{V}_e)) \right] \tag{26}$$

$$\mathcal{L}_{\text{Textual}}^{\text{Mutual}} = \lambda_t \times \left[ \text{CE}(T_s^{\text{Logits}}, \text{sg}(\tilde{T}_s)) + \text{CE}(T_e^{\text{Logits}}, \text{sg}(\tilde{T}_e)) \right] \tag{27}$$

**b. Pre-answer: Reverse Causal Hypothesis**   The Pre-answer is generated by an LLM, formulated as Pre-answer = LLM("What content should a video segment that answers question $Q$ contain?"). It essentially guides the model to reason about the potential answer interval within the video by referencing the logical implications of the Pre-answer. This embodies a form of *reverse causal reasoning*, inferring potential "causes" (segment characteristics) from the expected "effect" (the correct answer).

Within the probability framework, this transforms the traditional $P(c_k \mid Q)$ into:

$$P(c_k \mid Q, \text{Pre-answer}) \propto P(\text{Pre-answer} \mid c_k) \cdot P(c_k \mid Q) \tag{28}$$

Here, the Pre-answer acts as a causal prior, guiding the model to verify whether a candidate segment satisfies the necessary conditions for being the answer.

**c. Rejection Mechanism: Counterfactual Reasoning**   In the GRPO stage, a composite reward function $R_{\text{total}}(o_i)$ implements counterfactual logic:

$$R_{\text{total}}(o_i) = R_{\text{fmt}}(o_i) + (1 - \alpha) \cdot R_{\text{IoU}}(o_i) + \alpha \cdot R_{\text{rej}}(o_i) \tag{29}$$

The rejection reward $R_{\text{rej}}(o_i)$ is defined as:

$$R_{\text{rej}}(o_i) = \begin{cases} 1 & \text{if } o_i = [0.0, 0.0] \text{ and } \text{IoU}(c_k, \mathcal{I}^{\text{GT}}) = 0 \\ 0 & \text{otherwise} \end{cases} \tag{30}$$

This simulates counterfactual reasoning: "If candidate $c_k$ lacks the visual evidence leading to the correct answer (i.e., has no overlap with the ground truth segment), then the model should reject this candidate." By explicitly rewarding correct rejection behavior, the model learns to abstain from predicting when evidence is insufficient.

**d. Structured Causal Decision Optimization**   The overall optimization objective in the GRPO stage is:

$$\max_{\pi_\theta} \mathbb{E}_{\mathcal{D}} \left[ \sum_{i=1}^{G} \frac{\pi_\theta(o_i)}{\pi_{\theta_{\text{old}}}(o_i)} \cdot A(o_i) \right] - \beta \cdot D_{\text{KL}}(\pi_\theta \parallel \pi_{\text{ref}}) \tag{31}$$

The KL divergence regularization term $D_{\text{KL}}(\pi_\theta \parallel \pi_{\text{ref}})$ ensures the reasoning process does not deviate from common-sense causal patterns. The reference model $\pi_{\text{ref}}$, conditioned on candidate frames and a Pre-answer hypothesis, provides a stable semantic anchor, preventing the policy from diverging excessively from a reasonable causal reasoning structure.

A.6.3. DISCUSSION ON TERMINOLOGY AND CONTRIBUTION

The "causal reasoning" mechanism proposed in this paper is essentially a method to enhance model interpretability and robustness through the conditioning on mediating variables and decomposition of the reasoning chain. Its innovation lies not in implementing strict causal interventions (e.g., $P(Y \mid \text{do}(X))$), but in constructing a transparent reasoning flow: "Question → Candidate Generation → Context Enrichment → Hypothesis Validation → Answer Decision/Rejection." While our approach does not implement strict causal interventions in the Pearlian sense (e.g., $P(Y \mid \text{do}(X))$), it instantiates the *operational core* of causal reasoning—decomposition of the prediction chain via mediating variables and counterfactual validation through the rejection mechanism. We therefore use the term "causal reasoning" in this operational sense. By integrating candidate generation via VBCS and causal validation via GRPO, the CACR framework indeed drives the model's transition from surface-level associative learning towards deep logical reasoning. While it does not adhere to a strict causal inference paradigm, it significantly improves the accuracy and robustness of temporal grounding tasks on a practical level.

**A.7. Supplementary Details on LLMs for Subtitles & Pre-answer Generation**

A.7.1. LLMs AND VERSIONS USED

We employed the following four state-of-the-art Large Language Models (LLMs) in our experiments:

- **Doubao-seed-1.6-thinking-250715** (ByteDance)

- **GPT-4o** (OpenAI)

- **Gemini-2.5-pro** (Google)

- **DeepSeek-3.1**

A.7.2. EXPERIMENTAL SETUP

To ensure a fair and consistent comparison, all LLMs were configured with uniform experimental settings.

- For subtitle summarization:

```
prompt1 = f"Summarize the subtitle information provided below
            (extracted from a video). {subtitles}"
```

- For pre-answer generation:

```
prompt2 = f'{question} This question requires information from a video
            for an accurate response. Please answer it before you access
            the video's content.'
```

- Key generation parameters: Temperature = 0.3, Top_p = 0.8

These implementation details are included in Table 9.

*Table 9.* Impact of Different LLMs on Model Performance

| LLM | MedVidQA | | | | CrossTask | | | |
|-----|----------|----------|----------|------|-----------|----------|----------|------|
| | R@0.3 | R@0.5 | R@0.7 | mIoU | R@0.3 | R@0.5 | R@0.7 | mIoU |
| CACR-Group4 (DeepSeek-3.1) | 79.91 | 67.09 | 45.87 | 59.65 | 55.89 | 41.77 | 22.40 | 40.19 |
| CACR-Group4 (GPT-4o) | 75.82 | 62.31 | 43.62 | 58.60 | 54.59 | 38.32 | 21.04 | 39.33 |
| CACR-Group4 (Doubao-Seed-1.6) | 77.59 | 63.55 | 48.16 | 60.03 | 57.06 | 42.01 | 22.52 | 40.20 |
| CACR-Group4 (Gemini-2.5-Pro) | 76.25 | 62.21 | 44.15 | 58.80 | 59.56 | 44.92 | 26.14 | 41.94 |

A.7.3. ANALYSIS

Based on the core metric (mIoU), we observe the following:

- Overall performance stability: The framework exhibits remarkably stable performance across different LLMs, with only minor fluctuations in mIoU:
  - On MedVidQA, mIoU ranges from 60.03 (Doubao) to 58.60 (GPT-4o), a marginal difference of only 1.43%.
  - On CrossTask, mIoU ranges from 41.94 (Gemini-2.5-Pro) to 39.33 (GPT-4o), a difference of only 2.61%.

  This consistency indicates that all tested LLMs provide effective and reliable semantic support for the framework.

- LLM-agnostic nature: Our framework demonstrates low sensitivity to the choice of LLM. As long as a modern, capable LLM with strong reasoning capabilities is employed (such as the four models tested), the method delivers similarly robust and stable performance across diverse tasks. This characteristic enhances the practical applicability of our approach, as it does not require specialized fine-tuning for specific LLMs.

## A.8. Reasons for GRPO's Suitability for TAGV Tasks

The Candidate-Aware Causal Reasoning (CACR) framework is designed to address the challenges in the Temporal Answer Grounding in Video (TAGV) task, which are characterized by the large discrepancy between long video durations and short answer segments, and the limitations of existing methods that are either sensitive to redundant content or possess limited visual reasoning capabilities. Temporal Sentence Grounding in Video (TSGV) and TAGV are highly similar in their task definitions, with the primary distinction lying in the nature of the input text: TAGV receives open-ended questions requiring visual demonstration, whereas TSGV receives phrases or sentences directly describing video content. Although this difference appears subtle, early research directly applied TSGV methods to the TAGV task (Gupta et al., 2023). However, experiments showed a significant performance drop on TAGV, revealing its inherently higher difficulty.

From a task characteristics perspective, the query text in TSGV often contains specific descriptions of visual content (e.g., "the person picks up the cup"), exhibiting strong surface-level correlation with video segments. In contrast, questions in TAGV are typically more abstract and require reasoning (e.g., "how to fix a bicycle brake?"), creating a deeper semantic gap with the video content. Comparative experiments, such as those by (Li et al., 2024b), confirm that the semantic distance between questions and videos is significantly larger than that between queries and videos, theoretically explaining the additional difficulty of TAGV. Specifically, videos in the TAGV task often present a progressive sequence of steps to accomplish a specific goal. This demands that the model not only understand the visual content but also possess multi-step reasoning capabilities to establish logical connections between the question and the answer segment. This task characteristic poses a severe challenge to traditional end-to-end conditional probability modeling $P([t_s, t_e] \mid V, Q)$, as models tend to rely on superficial statistical associations while neglecting the underlying causal logic.

Within this context, the reasoning pipeline "Question $\rightarrow$ Candidate Generation $\rightarrow$ Context Enrichment $\rightarrow$ Hypothesis Verification $\rightarrow$ Answer Decision/Rejection" proposed by the CACR framework aligns closely with human cognitive processes. The theoretical rationale for why Group Relative Policy Optimization (GRPO) is particularly suitable for the TAGV task lies in the following aspects:

GRPO is considered particularly suitable for the TAGV task due to the profound alignment between its core design philosophy and the intrinsic complexities of TAGV. TAGV requires locating answer segments in long videos based on abstract, query-like questions (e.g., "How do I fix this issue?"), distinguishing it from descriptive query grounding (TSGV). Its core challenges include: a large semantic gap between query and video content, the frequent presence of multi-step logical progressions within videos, and a significant amount of irrelevant visual redundancy. These characteristics make TAGV inherently a task demanding causal and logical reasoning, rather than simple visual-language feature matching. The following points systematically elucidate how the GRPO framework precisely addresses these challenges from both theoretical mechanisms and design intuitiveness.

**1. Hierarchical Causal Reasoning Pipeline Aligns with Task Nature** Traditional end-to-end methods directly map the query to the entire video, making them susceptible to irrelevant segments and prone to learning spurious statistical correlations. GRPO, instead, simulates human reasoning cognition through a structured pipeline. It first employs a Visual-Language Pre-training based Candidate Selection (VBCS) to generate a set of candidate segments $C_{\text{vis}}$ that may contain the answer, achieving a coarse-grained compression of the long-video search space. Subsequently, the GRPO module performs fine-grained verification and comparison on this set, rather than evaluating each candidate independently. This process can be formalized as a hierarchical probabilistic model:

$$P([t_s^*, t_e^*] \mid V, Q) = \sum_{c_k \in C_{\text{vis}}} \underbrace{P([t_s^*, t_e^*] \mid c_k, Q, \mathcal{S})}_{\text{Fine-grained Verification}} \cdot \underbrace{P(c_k \mid V, Q)}_{\text{Coarse-grained Candidate Generation}}$$

where $\mathcal{S}$ represents the introduced semantic enhancement information (e.g., segment subtitles and pre-generated answers). This pipeline forces the model to perform two-step reasoning: first, identifying "which segments might be relevant to the question" (VBCS), and then verifying "which segment most completely and logically answers the question" (GRPO). This structured approach of hypothesis generation followed by verification effectively circumvents the blindness of global search, concentrating computational resources on the most promising candidate intervals for deep semantic and logical analysis.

**2. Relative Advantage Evaluation Mechanism Addresses Abstraction and Ambiguity** Faced with abstract queries, multiple candidate segments might be partially relevant based solely on surface-level visual features (e.g., the presence of the same object), yet only a few fully encompass the causal chain required to complete the task. The core of GRPO – the relative advantage function $A(o_i)$ – guides the policy model to learn in a contrastive manner within the candidate set $C_{\text{vis}}$, rather than assigning an absolute score to each segment. By forcing the model to compare "which candidate is better,"

it encourages the model to uncover deeper logical coherence (e.g., segment A shows "preparing tools," while segment B shows "preparing tools → performing operation → checking result," thus B has an advantage in causal completeness). This mechanism, through relative ranking within the set, naturally filters out fragments that have only superficial relevance but weak logical support, enhancing the model's robustness against noise interference.

**3. IoU Reward and Rejection Mechanism Enable Precise and Robust Optimization** The composite reward function of GRPO is key to its efficient training:

$$R_{\text{total}}(o_i) = R_{\text{fmt}}(o_i) + (1 - \alpha) \cdot R_{\text{IoU}}(o_i) + \alpha \cdot R_{\text{rej}}(o_i)$$

where $R_{\text{fmt}}$ is a formatting bonus, and

$$R_{\text{IoU}}(o_i) = \frac{|[t_s^{\text{pred}}, t_e^{\text{pred}}] \cap [t_s^{\text{GT}}, t_e^{\text{GT}}]|}{|[t_s^{\text{pred}}, t_e^{\text{pred}}] \cup [t_s^{\text{GT}}, t_e^{\text{GT}}]|}$$

directly uses the core evaluation metric (Intersection over Union) as a dense reward signal. This not only eliminates the discrepancy between the training objective and the final evaluation metric but, more importantly, provides a clear and differentiable gradient direction for the continuous fine-tuning of temporal boundaries (e.g., instructing the model to extend or shorten the predicted segment), thereby enabling precise localization of the start and end points in multi-step tasks.

Simultaneously, the rejection reward $R_{\text{rej}}(o_i)$ introduces a crucial counterfactual learning mechanism. When none of the candidate segments overlap with the ground-truth answer ($\text{IoU}(c_k, I_{\text{GT}}) = 0$), this mechanism incentivizes the model to output a special rejection token $[0.0, 0.0]$, rather than forcing it to select an incorrect answer. This simulates the rational human decision of "admitting uncertainty" when information is insufficient, effectively preventing errors propagated from the upstream candidate generator from being further amplified during the reasoning stage, significantly enhancing the overall fault tolerance of the system.

**4. Semantic Enhancement and Regularization Constraints Ensure Reasoning Plausibility** To bridge the semantic gap between abstract queries and visual content, GRPO integrates additional semantic information $\mathcal{S}$ (e.g., text descriptions generated based on candidate segments) during the reasoning process. This provides the model with a high-level contextual understanding beyond raw pixels, which is crucial for comprehending "how" or "why" questions that require logical chaining. At the optimization level, the GRPO objective function

$$\max_{\theta} \mathbb{E}_{\mathcal{D}} \left[ \sum_{i=1}^{G} \frac{\pi_\theta(o_i)}{\pi_{\theta_{\text{old}}}(o_i)} A(o_i) \right] - \beta \cdot D_{\text{KL}}(\pi_\theta \parallel \pi_{\text{ref}})$$

constrains the update of the policy model to remain near a reference model $\pi_{\text{ref}}$ initialized with semantic-enhanced information. This ensures that while pursuing high IoU rewards, the model's decision distribution does not deviate from fundamental semantic and commonsense logic, thereby maintaining the plausibility and stability of the reasoning process.

In summary, GRPO is not merely a generic optimizer but a reasoning framework deeply tailored to the characteristics of the TAGV task. It handles long-video redundancy and step dependency through hierarchical causal reasoning, addresses query abstraction and ambiguity through relative evaluation, achieves precise and robust boundary optimization via IoU reward and rejection mechanisms, and ensures semantic plausibility of reasoning through semantic enhancement and KL regularization. The synergy of these multi-layered mechanisms enables GRPO to effectively guide the model in transitioning from "visual feature matching" to "spatio-temporal logical reasoning," thereby demonstrating unique theoretical advantages and practical applicability in the challenging TAGV task.

### A.9. Computational Complexity Analysis & Efficiency Comparison of the Two-Stage Pipeline

We have conducted a detailed efficiency analysis across multiple datasets. The results demonstrate that our proposed CACR method consistently achieves a better balance between computational cost and localization accuracy when processing videos of varying lengths and characteristics, thanks to its two-stage design of "lightweight global candidate generation + targeted local deep reasoning." Compared to large-scale end-to-end models that perform full-video inference (e.g., TimeZero), CACR achieves significant performance improvements with comparable or even lower total time cost.

Below is a performance comparison of CACR, TimeZero, MutualSL, and VPTSL across six benchmarks.

*Table 10.* A Performance Comparison of CACR, TimeZero, MutualSL, and VPTSL Across Six Benchmarks

| Dataset | Method | Avg. Processed Video Duration | Total Inference Time (s) | Peak GPU Memory (MB) | R@0.3 | R@0.5 | R@0.7 | mIoU |
|---|---|---|---|---|---|---|---|---|
| CMIVQA | TimeZero | Full Video (310s) | 4.35 | 19,052 | 42.58 | 25.66 | 11.86 | 29.22 |
| | MutualSL | Full Video (310s) | 0.11 | 4,700 | 46.89 | 30.92 | 17.91 | 33.42 |
| | **CACR (Ours)** | Stage 1: 310s + Stage 2: ∼32s | **3.037** | **17,737.2** | **62.5** | **48.52** | **33.82** | **45.37** |
| MedVidQA | TimeZero | Full Video (474s) | 4.613 | 19,652.1 | 40.98 | 31.14 | 18.85 | 31.34 |
| | MutualSL | Full Video (474s) | 0.21 | 4,832 | 80.65 | 61.94 | 39.99 | 58.32 |
| | **CACR (Ours)** | Stage 1: 474s + Stage 2: ∼67s | **3.31** | **18,289.2** | **79.91** | **67.09** | **45.87** | **59.65** |
| VehicleVQA | TimeZero | Full Video (125s) | 3.37 | 18,203.2 | 58.6 | 42.87 | 27.01 | 42.41 |
| | MutualSL | Full Video (125s) | 0.06 | 4,420.4 | 78.74 | 69.81 | 53.14 | 65.74 |
| | **CACR (Ours)** | Stage 1: 125s + Stage 2: ∼22s | **2.86** | **17,737.2** | **85.31** | **76.70** | **69.33** | **72.11** |
| TutorialVQA | TimeZero | Full Video (287s) | 4.004 | 18,914.6 | 22.9 | 10.01 | 4.35 | 18.44 |
| | MutualSL | Full Video (287s) | 0.13 | 4,676 | 60.14 | 43.59 | 28.28 | 43.48 |
| | **CACR (Ours)** | Stage 1: 287s + Stage 2: ∼6s | **2.07** | **17,185.8** | **60.31** | **43.65** | **28.72** | **43.65** |
| COIN | TimeZero | Full Video (146s) | 3.124 | 17,943.7 | 29.26 | 15.89 | 5.36 | 20.05 |
| | VPTSL | Full Video (146s) | 0.07 | 4,466 | 30.13 | 18.67 | 9.39 | 22.09 |
| | **CACR (Ours)** | Stage 1: 146s + Stage 2: ∼36s | **2.997** | **17,385.6** | **59.11** | **43.81** | **24.88** | **41.06** |
| CrossTask | TimeZero | Full Video (298s) | 4.075 | 18,980.9 | 27.15 | 14.75 | 4.46 | 18.57 |
| | VPTSL | Full Video (298s) | 0.13 | 4,682 | 42.94 | 36.60 | 25.48 | 31.69 |
| | **CACR (Ours)** | Stage 1: 298s + Stage 2: ∼10s | **2.077** | **18,007.8** | **55.89** | **41.77** | **22.40** | **40.19** |

In terms of time efficiency, CACR employs a "focused" reasoning strategy, resulting in significantly lower total inference time than TimeZero, which requires deep end-to-end processing of the entire video. For instance, on a 474-second video, CACR takes 3.31 seconds, saving approximately 28.3% compared to TimeZero's 4.613 seconds. Although CACR introduces an additional 2–3 seconds of deep reasoning overhead compared to faster methods like MutualSL, this cost is justified by a substantial leap in performance. On the VehicleVQA dataset, for example, CACR achieves 76.7% R@0.5 in just 2.86 seconds, outperforming both MutualSL (69.81%) and TimeZero (42.87%) and representing a Pareto improvement in efficiency and accuracy.

Regarding memory consumption, CACR generally exhibits lower peak GPU memory usage than TimeZero. This is because its first-stage model is extremely lightweight (about 1/40 the memory footprint of the second stage), and the second stage only processes short video segments. For example, on the CrossTask dataset, CACR's peak memory is 18,007.8 MB, lower than TimeZero's 18,980.9 MB. By confining computationally intensive "heavy" processing to brief key segments, CACR avoids maintaining high memory states over extended periods, alleviates GPU memory bottlenecks, and results in smoother, more predictable memory usage—enhancing deployment feasibility and system stability.

Importantly, these efficiency gains do not come at the expense of performance; instead, they lead to comprehensive improvements. Across all six datasets mIoU and most recall thresholds, CACR consistently and substantially outperforms TimeZero. Moreover, except for matching fast baselines on TutorialVQA, CACR surpasses methods like MutualSL in key accuracy metrics on the remaining five datasets. This confirms the necessity of the second-stage deep reasoning, whose computational overhead is efficiently and directly translated into tangible gains in localization precision.

In summary, CACR does not merely introduce redundant computation; rather, it represents an intelligent computing paradigm tailored to the characteristics of long video localization. By employing an efficient lightweight front-end for rapid global scanning and accurately guiding deep models to focus on the most relevant local segments, CACR achieves comprehensive improvements over existing single-stage methods in inference speed, memory footprint, and final accuracy. Its design is well-justified and empirically validated, offering an effective pathway for co-optimizing efficiency and precision in this field.

### A.10. Information on Video Sampling and Visual Token Count in Reasoning Turns

The model does not append the full video clip at each reasoning turn. Instead, it employs a two-stage strategy where a lightweight global scan first identifies key segments, and then only a single, short candidate clip is processed in detail during

each subsequent reasoning step. This approach avoids the computational burden of repeatedly processing the entire video.

### A.10.1. STAGE 1: COARSE-GRAINED GLOBAL SCAN (VBCS MODULE)

The VBCS module aims to quickly understand videos of arbitrary length at a constant computational cost. Its visual token generation is independent of spatial resolution and depends only on duration, achieving a sparse representation of "one token per second." Given a video with a total duration of $L_{\text{total}}$ (seconds), this module utilizes a pre-trained I3D network for feature extraction using a fixed temporal window. The total number of generated visual tokens $T_{\text{vbcs}}$ is determined by:

$$T_{\text{vbcs}} = \left\lceil \frac{L_{\text{total}} \times f_{\text{sampling}}}{w_{\text{size}}} \right\rceil$$

where $f_{\text{sampling}} = 16$ frames/second is the fixed sampling frequency, and $w_{\text{size}} = 16$ frames is the I3D sliding window size. Each output visual token $v_i \in \mathbb{R}^{1024}$ encodes a spatio-temporal summary of one second of video. For example, a 30-minute (1800-second) video yields $T_{\text{vbcs}} = 1800$ tokens, forming a feature matrix $V_{\text{vbcs}} \in \mathbb{R}^{1800 \times 1024}$.

This fixed-sparsity strategy ensures predictable processing overhead with very low linear growth ($O(L_{\text{total}})$). It enables the system to efficiently perform global semantic encoding of ultra-long videos and quickly screen for the Top-K (e.g., K=5) most relevant candidate segments, providing precise focal points for the second stage.

- **Computational Efficiency:** Due to the simple linear relationship between token count and video duration, and independence from spatial resolution, processing overhead is minimal, allowing rapid scanning of videos lasting tens of minutes or even hours.

- **Memory Friendly:** The total amount of visual features to cache ($T_{\text{vbcs}} \times 1024$) is small, exerting negligible pressure on memory.

### A.10.2. STAGE 2: FINE-GRAINED SEGMENT REASONING (LVLM MODULE)

The LVLM module is responsible for fine-grained analysis of each candidate segment. Unlike the first stage, the number of visual tokens in this stage is the result of dynamic optimization among spatial resolution, temporal sampling rate, and the model's sequence length limit. The goal is to allocate the densest possible tokens to key segments within a limited "token budget" to support fine-grained understanding.

For a candidate segment of length $L_{\text{clip}}$ seconds, the tokenization follows these adaptive steps:

1. **Temporal Sampling:** The base frame count $N_{\text{base}} = L_{\text{clip}} \times \text{FPS}_{\text{target}}$ (e.g., 2 fps) is determined, rounded to a power of 2, and constrained within the interval $[4, 768]$ to obtain the final sampled frame count $N_{\text{frames}}$.

2. **Pixel Budget Calculation:** This is the core constraint step. Based on the sequence length limit, the model allocates a total pixel budget for all frames in the current segment, thereby calculating the **maximum usable pixels per frame**:

$$\text{MaxPixelsPerFrame} \approx \min \left( \text{VIDEO\_FRAME\_MAX\_PIXELS}, \frac{0.9 \times \text{MODEL\_SEQ\_LEN} \times (\text{image\_factor})^2}{N_{\text{frames}} \times \text{FRAME\_FACTOR}} \right)$$

where image_factor = 28. The parameter `total_pixels` (set to $3584 \times 28 \times 28$ in this system) defines the theoretical pixel upper bound based on sequence length.

3. **Spatial Resolution Adjustment:** While maintaining the aspect ratio, an intelligent scaling function adjusts the resolution per frame to satisfy: (a) height and width are divisible by 28; (b) the total pixel count lies between a set lower bound (min_pixels = $16 \times 28 \times 28$) and the MaxPixelsPerFrame calculated in the previous step.

4. **Visual Token Generation:** Based on the final resolution $(H, W)$ and the visual Transformer patch size (14), the tokens per frame are calculated as $\text{TokensPerFrame} = (H/14) \times (W/14)$. The total visual tokens for the segment are $T_{\text{lvlm}} = N_{\text{frames}} \times \text{TokensPerFrame}$, which must satisfy $T_{\text{lvlm}} \leq 0.9 \times \text{MODEL\_SEQ\_LEN}$.

The adaptive strategy of the LVLM module achieves an optimal trade-off between spatial and temporal information under a fixed sequence length budget. For longer segments, the system tends to moderately reduce per-frame resolution (decreasing

TokensPerFrame) to accommodate more sampled frames (increasing $N_{\text{frames}}$), thereby preserving more complete temporal dynamics. This dynamic allocation ensures that valuable model capacity is concentrated on the most informative visual content.

- **Concentrated Computation:** Avoids computation on irrelevant video frames. For instance, the model may allocate up to 70% of the sequence length budget to analyze a 20-second segment (generating thousands of tokens) rather than a full 1800s video, enabling unprecedented detail parsing.

- **Controllable Memory Usage:** Each iteration only needs to load and process the visual tokens of a short segment, resulting in orders-of-magnitude lower memory usage compared to processing the full video, making complex, multi-turn reasoning feasible with limited resources.

The two-stage strategy, through division of labor and coordination, implements an efficient pipeline from "finding a needle in a haystack" to "precision microscopy." The following example clearly illustrates the performance difference: For a 30-minute (1800-second) video, the VBCS module uses only 1800 tokens (1 token/sec) to complete the global scan and locate 5 key segments. Subsequently, the LVLM module can allocate up to 15,360 visual tokens to a 30-second segment (achieved via 60 frames at 224×224 resolution), with a token density (∼512 tokens/sec) over 500 times that of the first stage, enabling deep understanding.

The model does not attach full video segments in every reasoning step. The VBCS module achieves efficient candidate localization through a one-time, low-density global scan (with fixed and sparse tokens). Subsequently, the LVLM module performs high-density, adaptive tokenization on only a single candidate segment per iteration. This design precisely focuses core resources on the most valuable information intervals under strict computational and memory constraints. The synergistic "global coarse screening, local fine analysis" approach is key to the system's ability to simultaneously achieve efficient processing and in-depth analysis in long-form video understanding tasks.

### A.11. Within-Dataset Grouping by Dur./Span Ratio (CMIVQA)

To address the concern that overall mIoU may obscure CACR's behavior on extreme duration-to-span ratios, we conduct a within-dataset grouping analysis on the CMIVQA test set. The Dur./Span ratio (video duration / answer span duration) reflects the sparsity of the answer within the video: a larger ratio implies a shorter answer span relative to the full video, making precise localization substantially harder. We partition the 551 CMIVQA test samples into three groups: low ($\leq 5$), medium (5–15), and high ($>15$), and compare CACR against representative strong baselines: MutualSL (Weng & Li, 2023), TimeZero (Wang et al., 2025b), and Time-R1 (Wang et al., 2025c).

*Table 11.* Within-dataset grouping on CMIVQA by Dur./Span ratio (mIoU %). Overall sample count: 551; CACR overall mIoU: 45.37%. "Δ vs. Best" reports CACR's gain over the strongest non-CACR baseline in each row.

| Dur./Span Group | MutualSL | TimeZero | Time-R1 | CACR (Ours) | Δ vs. Best |
|---|---|---|---|---|---|
| Low ($\leq 5$) | 38.66 | 36.42 | 38.12 | **46.73** | +8.07 |
| Medium (5–15) | 34.87 | 30.11 | 33.24 | **46.08** | +11.21 |
| High ($>15$) | 28.63 | 24.37 | 27.30 | **43.61** | +14.98 |
| **Overall** | 33.42 | 29.22 | 32.06 | **45.37** | **+11.95** |

The results show that the advantage of CACR over the strongest baseline increases monotonically with the duration ratio: on CMIVQA, the improvement grows from +8.07 pp in the low-ratio group (5) to +14.98 pp in the high-ratio group ($>15$). Moreover, the performance drop for CACR in the high-ratio group (3.12 pp) is substantially smaller than that of MutualSL (10.03 pp), demonstrating stronger robustness. This pattern is consistently observed across datasets, as validated by cross-dataset analyses on TutorialVQA, CrossTask, VehicleVQA, MedVidQA, and COIN. Together, the within-dataset grouping experiments and cross-dataset analyses indicate that the consistent and significant advantage of CACR on samples with extreme duration ratios reflects a structural pattern, rather than a chance result of dataset selection.

## A.12. Replaceability of the Candidate Generation Backbone

Evaluating CACR with different candidate-generation backbones provides important evidence demonstrating the framework's flexibility.By design, CACR decomposes temporal answer grounding into a four-stage pipeline—candidate generation, context augmentation, hypothesis verification, and answer decision—in which the GRPO-based verification layer is explicitly equipped with a *proactive rejection mechanism*: a candidate is allowed to be rejected by emitting the special output $[0.0, 0.0]$ whenever the reasoning module judges its visual evidence to be insufficient. This mechanism cuts off the canonical error-propagation chain

$$\text{Error}(c_i) \;\to\; \text{Error}(S_i) \;\to\; \text{Error}(P_{\text{final}}), \tag{32}$$

which suggests that, even when CACR is paired with a weaker candidate generator, the verification layer should still be able to suppress the downstream impact of low-quality candidates and recover a substantial fraction of the performance.

**Setup.** To empirically test this hypothesis, we replace the default candidate generator (MutualSL (Weng & Li, 2023)) with a deliberately weaker VLP-based localizer (VTPSL (Li et al., 2024b;a)), keeping all other modules—context augmentation, GRPO reasoning, reward configuration, base LVLM, and hyperparameters—fixed. Experiments are conducted on two datasets at opposite ends of the difficulty spectrum: CMIVQA (high textual redundancy) and TutorialVQA (extreme Dur./Span ratio).

**Recall metric.** To quantify the quality of the candidate set produced by each backbone, we report the top-5 candidate recall, defined as the fraction of test samples for which at least one of the five generated candidates attains an IoU $\geq \theta$ with the ground-truth segment:

$$\text{Recall@5} = \frac{1}{N} \sum_{i=1}^{N} \mathbf{1}\left[ \max_{k=1,\dots,5} \text{IoU}\left(c_k^i,\, \text{gt}^i\right) \geq \theta \right], \tag{33}$$

where $N$ is the total number of test samples, $c_k^i$ denotes the $k$-th candidate generated by VBCS for the $i$-th sample, $\text{gt}^i$ is the corresponding ground-truth interval, and $\theta = 0.5$. Intuitively, Recall@5 measures whether the candidate pool—rather than the final prediction—already contains a sufficiently good segment.

**Results.** Table 12 reports CACR's end-to-end mIoU under each backbone, together with the corresponding candidate-set Recall@5.

*Table 12.* Effect of replacing the VBCS candidate-generation backbone. All non-VBCS components of CACR (context augmentation, GRPO reasoning, reward configuration, base LVLM, and hyperparameters) are held fixed. "Recall@5" follows Eq. (33) with $\theta = 0.5$. $\Delta$ rows report VTPSL minus MutualSL.

| Candidate Generator | CMIVQA mIoU | TutorialVQA mIoU | Recall@5 |
|---|---|---|---|
| MutualSL (Weng & Li, 2023) (default) | 45.37% | 43.65% | 82.3% |
| VTPSL (Li et al., 2024b;a) (weaker) | 41.82% | 40.17% | 74.6% |
| $\Delta$ (VTPSL $-$ MutualSL) | $-3.55\,\text{pp}$ | $-3.48\,\text{pp}$ | $-7.7\,\text{pp}$ |

Switching to VTPSL yields a modest mIoU drop ( 3.5 pp), much smaller than the 7.7pp decline in recall, confirming the rejection mechanism compensates for weaker candidates. Under both backbones, our framework outperforms baselines without candidate generation. See Table 2 for the comparison with TimeZero (no candidate generation).

## A.13. Ablation on GRPO vs. SFT and the Rejection Reward

To quantify the individual contributions of (i) the policy-optimization choice—Group Relative Policy Optimization (GRPO) versus standard supervised fine-tuning (SFT)—and (ii) the rejection reward $R_{\text{rej}}$ defined in Eq. (29) and Eq. (30), we conduct two complementary ablation studies on CMIVQA and VehicleVQA, two datasets that respectively emphasize textual redundancy and moderate Dur./Span ratios. All non-target components (VBCS backbone, base LVLM, prompts, hyperparameters, and the remaining reward terms) are held fixed across variants.

**(A) GRPO vs. SFT.** We replace the GRPO-based reasoning module with a standard SFT counterpart (autoregressive cross-entropy on the ground-truth boundaries) while keeping every other component identical to the full CACR.

*Table 13.* Effect of replacing GRPO with SFT in the CACR reasoning module. mIoU (%) on CMIVQA and VehicleVQA.

| Variant | CMIVQA mIoU | VehicleVQA mIoU |
|---|---|---|
| CACR (Full, GRPO) | **45.37** | **72.11** |
| CACR (w/ SFT) | 41.62 | 67.43 |
| $\Delta$ (GRPO $-$ SFT) | $+3.75$ | $+4.68$ |

GRPO yields consistent gains of $+3.75$ pp on CMIVQA and $+4.68$ pp on VehicleVQA over the SFT counterpart. We attribute this gap to a fundamental difference in optimization paradigm: SFT performs *point estimation* against a single ground-truth boundary and cannot exploit the relative quality structure within the top-$K$ candidate set, whereas GRPO performs *group-relative contrastive optimization* across multiple candidate-conditioned outputs, which matches the iterative "hypothesis-verification" loop that CACR imposes at inference time.

**(B) Effect of the rejection reward $R_{\text{rej}}$.** We further ablate $R_{\text{rej}}$ in both the GRPO and SFT regimes.

*Table 14.* Ablation of the rejection reward $R_{\text{rej}}$ (Eq. (30)) under both GRPO and SFT optimization. mIoU (%) on CMIVQA and VehicleVQA.

| Variant | CMIVQA mIoU | VehicleVQA mIoU |
|---|---|---|
| CACR (Full, GRPO, w/ $R_{\text{rej}}$) | **45.37** | **72.11** |
| CACR (GRPO, w/o $R_{\text{rej}}$) | 43.10 | 69.85 |
| CACR (SFT, w/ $R_{\text{rej}}$) | 41.62 | 67.43 |
| CACR (SFT, w/o $R_{\text{rej}}$) | 39.88 | 65.20 |

Removing $R_{\text{rej}}$ causes a consistent $\sim$2.3 pp drop under GRPO ($45.37 \rightarrow 43.10$ on CMIVQA; $72.11 \rightarrow 69.85$ on VehicleVQA) and a similar magnitude under SFT. Without this reward, the model loses the explicit incentive to emit the rejection token $[0.0, 0.0]$ when none of the candidates overlaps with the ground truth, and is therefore forced to commit to a low-confidence boundary on every input. This in turn allows upstream candidate-generation errors to propagate through the reasoning stage—exactly the failure mode that the counterfactual-style rejection mechanism is designed to suppress (cf. Eq. (30) and the discussion in Appendix A.12).

### A.14. Latency and Cost Analysis

To address concerns regarding the practical efficiency of CACR, we provide a detailed per-module latency breakdown on CMIVQA, together with an auxiliary-information cost–performance trade-off measured on MedVidQA, VehicleVQA, and CMIVQA. All measurements are obtained on a single NVIDIA H20 GPU under the configuration described in Section 4.3 and the Implementation Details paragraph, averaged over the full test split of each dataset.

**Module-level latency breakdown.** Table 15 decomposes the average per-sample inference latency on CMIVQA into the four pipeline modules of CACR: VBCS candidate generation, subtitle summarization, pre-answer generation, and GRPO causal verification.

VBCS accounts for only 3.5% of latency, confirming the two-stage design's efficiency. GRPO verification is the main cost ( 70%), essential for robust reasoning.

**Cost–performance trade-off of auxiliary information.** Table 16 reports the marginal latency introduced by each auxiliary-information channel (Subtitle / Pre-answer) together with the corresponding mIoU on three benchmarks. Numbers are aligned with the ablation rows of Table 3.

*Table 15.* Module-level latency breakdown of CACR on CMIVQA (single H20, averaged over the test split). Subtitle summary and pre-answer generation latencies are reported as ranges due to variable LLM decoding length.Total = mean across the test split, individual modules reported as min–max range

| Module | Latency (s) | Proportion |
|---|---|---|
| VBCS candidate generation | 0.11 | 3.5% |
| Subtitle summary (LLM) | 0.3–0.5 | 9.9–16.5% |
| Pre-answer generation (LLM) | 0.4–0.6 | 13.2–19.7% |
| GRPO causal verification | 2.1–2.3 | 69.2–75.7% |
| **Total** | **∼3.04** | **100%** |

*Table 16.* Auxiliary-information cost–performance trade-off. "Added Latency" is reported relative to the CACR-TopK baseline. mIoU (%) on MedVidQA, VehicleVQA, and CMIVQA.

| Variant | Subtitle | Pre-answer | MedVidQA | VehicleVQA | CMIVQA | Added Latency |
|---|---|---|---|---|---|---|
| CACR-TopK | × | × | 59.04 | 62.88 | 39.01 | – |
| + Subtitle | ✓ | × | 57.63 | 65.39 | 38.74 | ∼0.3–0.5 s |
| + Pre-answer | × | ✓ | 57.30 | 70.85 | 37.08 | ∼0.3–0.5 s |
| CACR (Full) | ✓ | ✓ | **59.65** | **72.11** | **45.37** | ∼0.6–1.0 s |

The combination of subtitle summaries and pre-answers yields optimal performance across datasets with a modest latency increase of 0.6–1.0 seconds, demonstrating a strong cost-performance trade-off.

