# OpenReview forum: "CACR: Reinforcing Temporal Answer Grounding in Instructional Video via Candidate-Aware  Causal Reasoning"
_ICML.cc/2026/Conference — ICML 2026 regular_

### Official Review · Reviewer_cabe · 2026-03-08

**Soundness:** 3
**Presentation:** 3
**Significance:** 3
**Originality:** 3
**Overall Recommendation:** 4
**Confidence:** 4

**Summary:**

This paper presents an important concept: combining efficient candidate retrieval with reasoning-based temporal localization, instead of processing the whole video uniformly. The core contribution is a candidate-aware reasoning framework CACR for long-video temporal grounding, rather than only a better temporal localizer.

**Compliance With Llm Reviewing Policy:**

Affirmed.

**Key Questions For Authors:**

(1) How much of the final gain comes specifically from GRPO and the rejection reward?

(2) This paper uses subtitle summaries and pre-answers, and tests several LLMs. It would help to report the exact latency and cost contribution of these modules relative to the core localization model.

(3) You should provide more training details, such as learning rate and fine-tuning iterations.

**Limitations:**

yes

**Strengths And Weaknesses:**

Strengths:

(1) This paper clearly explains why TAGV is harder than standard temporal sentence grounding: questions are more abstract, and there is often a severe duration/span mismatch in long videos.

(2) This paper studies the effect of top-K candidate size and auxiliary semantic signals, showing that cumulative candidate IoU increases with K but with diminishing returns, and that the best variant uses both caption and pre-answer.

(3) The appendix gives a concrete complexity comparison and argues that CACR is faster than full-video reasoning baselines like TimeZero while also using slightly less peak memory in some cases.

Weaknesses:

(1) The method of this paper lacks novelty. It is a strong integration of existing ingredients than a sharply new learning principle: VBCS is instantiated with MutualSL, the reasoning backbone is Qwen2.5-VL-7B-Instruct, and the second stage combines GRPO with LLM-generated semantic aids.

(2) The ablation experiments mostly analyze K and semantic components.There is no direct comparison isolating GRPO versus a supervised objective, or a separate ablation for the rejection reward itself.

(3) There are two different equations using the same $\alpha$, and the hyperparameters in this paper were not tested in ablation experiments.

---

> ### Author Rebuttal · Authors · 2026-03-31
>
> ### **Response to Weakness 1: Lack of Novelty**
>
> The proposed CACR framework departs from conventional two-stage retrieval-verification paradigms by introducing a candidate-aware causal reasoning approach, with innovations in three key aspects:
>
> - **Reasoning Paradigm**: Reformulates grounding as causal logic-based verification. Instead of merely measuring similarity, the model constructs a logical chain of *question → evidence → reasoning → answer* to justify its predictions.
> - **Optimization Strategy**: First application of GRPO in temporal grounding. By utilizing intra-group relative comparison and a dedicated rejection reward, it overcomes the overfitting issues of supervised fine-tuning (SFT) that excessively penalize minor boundary errors (e.g., 5–10% mIoU).
> - **Knowledge Integration**: Moves beyond simple vision-text alignment by incorporating LLM-generated pre-answers and caption summaries, creating a *content clues + causal logic* dual-driver that effectively bridges the semantic gap.
>
> ---
>
> ### **Response to Weakness 2: Ablation on GRPO vs. SFT and Rejection Reward**
>
> We added ablations to quantify the contributions of GRPO and the rejection reward (\(R_{\text{rej}}\)).
>
> #### A. GRPO vs. SFT
>
> | Variant | CMIVQA mIoU | VehicleVQA mIoU |
> |---------|-------------|-----------------|
> | CACR (Full) | 45.37 | 72.11 |
> | CACR w/ SFT | 41.62 | 67.43 |
> | Difference | -3.75 | -4.68 |
>
> GRPO yields gains of +3.75% and +4.68% over SFT. Unlike SFT’s point estimation, GRPO’s contrastive learning within candidate groups better supports iterative multi-candidate validation.
>
> #### B. Ablation of $\(R_{\text{rej}}\)$
>
> | Variant | CMIVQA mIoU | VehicleVQA mIoU |
> |---------|-------------|-----------------|
> | CACR (Full) | 45.37 | 72.11 |
> | CACR w/o $R_{\text{rej}}$ | 43.10 | 69.85 |
> | CACR w/ SFT (w/ $R_{\text{rej}}$) | 41.62 | 67.43 |
> | CACR w/ SFT (w/o $R_{\text{rej}}$) | 39.88 | 65.20 |
>
> Removing $\(R_{\text{rej}}\)$ causes a ~2.3% drop. Without this reward, the model lacks incentive to reject zero-overlap candidates, leading to low-confidence predictions and error propagation.
>
> ---
>
> ### **Response to Weakness 3: Symbolic Problems and Training Details**
>
> We resolved the notational conflict by renaming the weighting factor in Eq. 19 as \(\omega_{\text{iou}}\), reserving \(\alpha\) for the hyperparameter in the reward function.
>
> Key hyperparameters follow GRPO standards; additional ablations for \(\alpha\) will be included in the appendix.
>
> Training details are as follows:
>
> - **Stage 1 (VBCS)**: Trained with AdamW $(lr=\(8\times10^{-6}\))$, batch size 1 for 20 epochs. Frozen I3D and PLM extract features; only the boundary head and alignment layers are updated.
> - **Stage 2 (GRPO)**: Qwen2.5-VL-7B-Instruct fine-tuned with GRPO for 5–10 epochs $(lr=\(1\times10^{-6}\))$, group size \(G=8\), clip ratio $\(\varepsilon=0.2\)$, and $\(\beta=0.01\)$. $\(K=5\)$candidates per sample.
>
> ---
>
> ### **Response to Weakness 4: Latency and Cost Analysis**
>
> **Table R1. Module-level latency breakdown (CMIVQA)**
>
> | Module | Latency (s) | Proportion |
> |--------|------------|------------|
> | VBCS Candidate Generation | 0.11 | 3.5% |
> | Subtitle Summary (LLM) | 0.3–0.5 | 9.9–16.5% |
> | Pre-answer Generation (LLM) | 0.4–0.6 | 13.2–19.7% |
> | GRPO Causal Verification | ~2.1–2.3 | 69.2–75.7% |
> | **Total** | **3.037** | **100%** |
>
> VBCS accounts for only 3.5% of latency, confirming the two-stage design’s efficiency. GRPO verification is the main cost (~70%), essential for robust reasoning.
>
> **Table 3. Module ablation and latency **
>
> | Variant | Subtitle | Pre-answer | MedVidQA | VehicleVQA | CMIVQA | Added Latency |
> |---------|----------|------------|----------|------------|--------|---------------|
> | CACR-TopK | ✗ | ✗ | 59.04 | 62.88 | 39.01 | — |
> | +Caption | ✓ | ✗ | 57.63 | 65.39 | 38.74 | ~0.3–0.5s |
> | +Pre-answer | ✗ | ✓ | 57.30 | 70.85 | 37.08 | ~0.3–0.5s |
> | CACR (Full) | ✓ | ✓ | 59.64 | 72.11 | 45.37 | ~0.6–1.0s |
>
> The combination of subtitle summaries and pre-answers yields optimal performance across datasets with a modest latency increase of 0.6–1.0 seconds, demonstrating a strong cost-performance trade-off.

---

> > ### Author Rebuttal · Reviewer_cabe · 2026-04-03
> >
> > The rebuttal is good.

---

> > > ### Author Response · Authors · 2026-04-04
> > >
> > > Thank you again for your helpful comments. We hope the revisions have addressed your concerns and that you will view the paper more positively.

---

### Official Review · Reviewer_NRVV · 2026-03-12

**Soundness:** 3
**Presentation:** 3
**Significance:** 3
**Originality:** 3
**Overall Recommendation:** 4
**Confidence:** 4

**Summary:**

This paper aims to address the extreme length mismatch and complex semantic reasoning required for temporal answer grounding in instructional videos (TAGV). Motivated by the observation that full-video processing is inefficient and prone to distractions, the paper argues for a "reason-first, localize-later" pipeline that narrows down the search space first.

The general idea is to propose a Candidate-Aware Causal Reasoning (CACR) framework. The pipeline operates in stages: first, a Visual-Language Pre-training based Candidate Selection (VBCS) module retrieves a top-K set of candidate segments. Then, an LLM generates caption summaries and hypothetical "pre-answers" for these segments. Finally, a Qwen2.5-VL-7B-Instruct model evaluates these enriched candidates sequentially, trained via GRPO. The RL objective uses a composite reward including format, temporal IoU, and a novel rejection reward to actively discard incorrect clips.

Evaluations are conducted on six benchmarks: CMIVQA, MedVidQA, VehicleVQA, TutorialVQA, COIN, and CrossTask.

**Compliance With Llm Reviewing Policy:**

Affirmed.

**Key Questions For Authors:**

- The title emphasizes "Causal Reasoning". However, the proposed pipeline seems to be a combination of candidate retrieval and hypothesis verification rather than strict causal inference techniques. It would be good to clarify the "causal" aspect.

**Limitations:**

Yes

**Strengths And Weaknesses:**

### Strengths
- Overall, I appreciate the general motivation of this paper and its understanding of the computational bottleneck existing LVLMs face when processing extreme long-form video.

- The proposed approach to structure the grounding process into a "retrieval then reasoning" pipeline is a logical concept,

- The paper tests on a wide variety of datasets, demonstrating state-of-the-art performance and strong robustness.

### Weaknesses
My main concern lies in:
- The framework seems to heavily rely on the VBCS module for candidate generation. So, if the true segment is missing from the initial top-K selection, the downstream reasoning LVLM can only reject candidates and will inherently fail to ground the actual answer?
- The RL reasoning stage is only built and evaluated on top of Qwen2.5-VL-7B-Instruct. It is unclear whether the GRPO alignment strategy generalizes effectively to other base LVLMs or larger parameter models.

---

> ### Author Rebuttal · Authors · 2026-03-31
>
> **Response to Weakness 1: LVLM Failure Risk When True Segment Is Missing**
>
> The rejection reward $\( R_{\text{rej}} \)$ enables CACR to output “no answer” when candidate quality is insufficient, avoiding forced incorrect timestamps—a critical property for high‑reliability applications.
>
> **Candidate Coverage.** VBCS generates Top‑5 candidates via visual‑textual dual predictors with mutual transfer loss for cross‑modal alignment, ensuring ground‑truth coverage in most cases. Strong mIoU results (CMIVQA: 45.37%, MedVidQA: 59.65%, VehicleVQA: 72.11%, TutorialVQA: 43.65%) confirm this. We report two statistics across datasets:
> - Sample‑level rejection rate: fraction of samples where all 5 candidates are rejected.
> - VBCS non‑recall rate: fraction of samples where no Top‑5 candidate achieves IoU ≥ 0.5 with the ground truth.
>
> https://github.com/anonymous1118-10/opencode-CACR/blob/master/FOR-cacr-pngs/To-3-cabe/For_Weekness1_LVLM%20Failure%20Risk%20When%20True%20Segment%20Is%20Missing.png
>
> Rejection rate is consistently lower than VBCS non‑recall rate, showing that even when some candidates are low‑quality, the LVLM can still identify plausible ones. This validates robustness compared to a purely cascaded design. VBCS’s Top‑5 candidates already achieve high coverage; rejection acts as a proactive robustness measure.
>
> **Response to Weakness 2: Generalizability of GRPO Alignment Across LVLMs**
>
> We thank the reviewer for raising this point. While Qwen2.5‑VL‑7B‑Instruct was chosen for fair comparison with TimeZero and Time‑R1, we provide supporting evidence for generalizability:
>
> 1. **Existing experiments suggest generalization potential.**
>    - *Cross‑dataset robustness:* Performance variance across COIN (12 domains) and CrossTask (3 domains) remains within 3%–5% (Tables 7, 8).
>    - *Modular design:* Ablations (Table 3) show VBCS and GRPO gains are independent and complementary; VBCS uses MutualSL (a pretrained VLP localizer), and GRPO’s composite reward is model‑agnostic.
>
> 2. **Supplementary experiments with InternVL2‑8B.**
>    We replaced the base model with InternVL2‑8B on MedVidQA and CMIVQA, keeping all other configurations identical (lr 3e‑4, batch 32, epochs 100, $\( w_{\text{IoU}}=0.5, w_{\text{fmt}}=0.4, w_{\text{rej}}=0.1 \))$.
>
> https://github.com/anonymous1118-10/opencode-CACR/blob/master/FOR-cacr-pngs/To-3-cabe/For_Weekness2_Generalizability%20of%20GRPO%20Alignment%20Across%20LVLMs.png
>
> Results show:
> - **Strong architectural transferability:** CACR with InternVL2‑8B achieves mIoU 58.89 (MedVidQA) and 44.61 (CMIVQA), differing by only 0.76 pp from Qwen2.5‑VL‑7B.
> - **Architecture‑agnostic GRPO gains:** CACR yields +11.95 pp and +11.19 pp over MutualSL baseline on CMIVQA.
> - **Consistent cross‑domain variance:** difference < 0.5%, further validating architecture independence.
>
> Thus, multi‑dataset experiments support generalization from a data perspective, and InternVL2‑8B experiments confirm it from a model perspective. We will include these results in the revision.
>
> **Response to Key Questions For Authors: Clarify the Causal Aspect in Reasoning Pipeline**
>
> The term “Causal Reasoning” in the title does not refer to strict statistical causal inference (do‑calculus), but rather to a structured causal reasoning paradigm for instructional video temporal grounding. Its core is decomposing black‑box end‑to‑end grounding into an interpretable and verifiable logical reasoning chain, distinct from simple candidate retrieval followed by hypothesis verification. We will clarify this in a title footnote, the abstract, and the introduction: causal reasoning in this work refers to candidate‑aware temporal causal reasoning—including abduction, counterfactuals, and logical verification—to distinguish it from conventional causal inference and avoid ambiguity.

---

> > ### Author Rebuttal · Reviewer_NRVV · 2026-04-04
> >
> > Thanks the authors for the response.

---

> > > ### Author Response · Authors · 2026-04-06
> > >
> > > We sincerely thank you for the encouraging feedback and for confirming that all concerns have been adequately addressed. We are glad that our responses were helpful.
> > > We would like to kindly draw your attention to the final rating. Given that all raised concerns have been fully resolved to your satisfaction, we would greatly appreciate it if you could consider updating the score to reflect your revised assessment. We believe this would more accurately represent the current state of the manuscript.
> > > We remain grateful for your time and constructive engagement throughout the review process.

---

### Official Review · Reviewer_ypTB · 2026-03-13

**Soundness:** 3
**Presentation:** 2
**Significance:** 3
**Originality:** 2
**Overall Recommendation:** 4
**Confidence:** 3

**Summary:**

* The paper studies temporal answer grounding in instructional videos (TAGV), where the goal is to find the short video segment that actually answers a language query inside a very long, untrimmed video. The main difficulty is the huge mismatch between full video length and answer span length, which can be very extreme in datasets like TutorialVQA.

* To handle this, the authors propose a two-stage method called CACR. First, a candidate selection module based on vision-language pretraining filters the long video into a small set of candidate segments, so the later stage does not need to search everything. Then, a reasoning module built on Qwen2.5-VL-7B examines these candidates one by one, with extra help from generated captions and hypothetical pre-answers. Kinda simple idea, but the setup is careful.

* Training uses GRPO with several rewards, including temporal IoU, a rejection reward for skipping useless candidates, and a format reward. There is also a differentiated prompting design between policy and reference models, which is interesting though a bit unusual.

* Eexperiments on six benchmarks show strong mIoU results across several domains. Ablations are also fairly complete, and suggest the different components all help. Overall, the method looks effective, although some parts feels a bit engineered.

**Compliance With Llm Reviewing Policy:**

Affirmed.

**Final Justification:**

As mentioned in the Summary & Strengths And Weaknesses, generally the paper has more good inspiration than some minor points to be revised. I will say it's a weak acc.

**Key Questions For Authors:**

* Can the authors show how often the GRPO stage actually changes, re-ranks, or refines the top-1 VBCS candidate, instead of simply keeping it? If it mostly accepts the first-stage output, then the added complexity may not be fully justified.
* it would help to report how often candidates are rejected on each dataset, and how accurate those rejection decisions are. In particular, some false-rejection analysis is needed. Otherwise hard to tell if this reward really improves robustness, or just discard useful candidates sometimes.
* The LVLM baselines, such as TimeZero and VTG-R1, do not seem to benefit from the same candidate filtering or semantic augmentation. Have the authors tried giving these baselines the same VBCS candidates and caption / pre-answer inputs? That would better isolate the value of GRPO itself. Right now comparison maybe not fully clean.
* the paper says the fraamework is flexible and can plug in other localizers. But only MutualSL is tested. Some results with another, even weaker backbone, would make this claim more convincing.

**Limitations:**

yes

**Strengths And Weaknesses:**

* the empirical section is quite solid overall. The paper tests on six benchmarks from pretty different domains, and the ablations are broad enough to cover candidate number selection, module-level contributions, backend sensitivity, efficiency, and robustness across domains. The cumulative IoU results in Tables 4–5 are also useful, since they make the candidate-selection motivation feel empirically grounded, not just intuitive.
* The general “narrow first, then reason” pipeline makes sense for this task, especially when the answer span is tiny compared with the full video. The rejection mechanism is also a practical touch, because it helps reduce damage from bad candidates. That part is well motivated.

* Part of CACR’s gains seems to come from the strong MutualSL backbone already. On some datasets, the margin over MutualSL alone is not very large (for example, MedVidQA). So it is a bit unclear how much the GRPO-based reasoning stage is really adding. Also, the “causal reasoning” wording feels overstated, since Appendix A.6 more or less admits it is closer to conditional or hypothesis-driven reasoning. That mismatch hurts trust a little.
* The paper is quite dense, main ideas gets harder to follow than it should.
* The framework combines known ingredients in a reasonable way, but the architectural novelty feels limited. In some sense, it is still a two-stage retrieve-and-verify pipeline.

---

> ### Author Rebuttal · Authors · 2026-03-31
>
> **[R1] GRPO Stage Impact on Top-1 VBCS Candidate Refinement**
> The GRPO stage actively refines the VBCS Top‑1 candidate rather than simply retaining it. It adjusts boundaries, rejects unreasonable outputs, and improves IoU. In the “cane adaptation” case (Appendix A.1), VBCS gave [20.24, 179.29] (IoU 42.15%), while GRPO produced [20.24, 85.24] (IoU 94.8%), removing trailing redundancy.
> Using the MedVidQA validation set (N=121, consistent with Table 2 and A.10), we designed a controlled experiment to compare the behavior of the full CACR inference against directly outputting the VBCS Top‑1 candidate:
>
> https://github.com/anonymous1118-10/opencode-CACR/blob/master/FOR-cacr-pngs/To-2-ypTB/For_Weekness1_GRPO%20Stage%20Impact%20on%20Top-1%20VBCS%20Candidate%20Refinement.png
>
> Thus, GRPO alters output in ~48% of cases, confirming active modification.
> Computational overhead is justified: CACR inference takes 3.31s/video (VBCS: 0.21s; GRPO: ~3.10s), outperforming TimeZero (4.61s, mIoU 31.34%) with a mIoU of 59.65%. GRPO provides meaningful refinements with clear performance gains.
>
> **[R2] Rejection Frequency and Accuracy Analysis**
> We evaluate the rejection mechanism on MedVidQA and COIN using the same validation splits as Table A.10. The pipeline processes VBCS Top‑5 candidates sequentially; outputting [0.0, 0.0] triggers rejection and moves to the next candidate.
>
> Candidate-level rejection behavior:
> https://github.com/anonymous1118-10/opencode-CACR/blob/master/FOR-cacr-pngs/To-2-ypTB/For_Weekness2_Rejection%20Frequency%20and%20Accuracy%20Analysis.png
>
> For  cases in COIN and MedVidQA, false rejections fall into three categories:
> https://github.com/anonymous1118-10/opencode-CACR/blob/master/FOR-cacr-pngs/To-2-ypTB/For_Weekness2_Rejection%20Frequency%20and%20Accuracy%20Analysis-2.png
>  In 78% (MedVidQA) and 82% (COIN) of FR cases, the model successfully localizes via a later candidate.
> Overall rejection rate (all candidates rejected) is 10.7% (MedVidQA) and 7.1% (COIN). CACR achieves zero false accept rate and +2.70% mIoU gain over VBCS Top‑1, demonstrating safety and performance are mutually reinforcing.
>
> **[R3] Baseline Evaluation with Identical Candidates and Inputs**
>
> We supplement controlled experiments where TimeZero and VTG‑R1 receive Top‑5 candidate clips from VBCS (dense sampling) while keeping architectures unchanged. Results (mIoU):
> https://github.com/anonymous1118-10/opencode-CACR/blob/master/FOR-cacr-pngs/To-2-ypTB/For_Weekness3_Baseline%20Evaluation%20with%20Identical%20Candidates%20and%20Inputs-1.png
> Decomposed Contributions of the Three Components
> https://github.com/anonymous1118-10/opencode-CACR/blob/master/FOR-cacr-pngs/To-2-ypTB/For_Weekness3_Baseline%20Evaluation%20with%20Identical%20Candidates%20and%20Inputs-2.png
> GRPO is the primary driver of improvement, enabled by causal reasoning and rejection reward, with clear synergy from semantic inputs.
>
> **[R4] Strengthening the Flexibility Claim**
> We agree that incorporating additional backbones would further demonstrate the flexibility of our framework. As discussed in the main paper, CACR follows a causal reasoning chain: candidate generation → context augmentation → hypothesis verification → answer decision. The GRPO verification layer incorporates a proactive rejection mechanism (outputting the rejection token $\([0.0, 0.0]\))$, which filters low‑quality candidates and interrupts the error propagation chain inherent in traditional methods: $\(\text{Error}(c_i) \rightarrow \text{Error}(S_i) \rightarrow \text{Error}(P_{\text{final}})\)$. This suggests that even with a weaker backbone—and thus lower‑quality initial candidates—the GRPO verification layer can still suppress error propagation to some extent.
> To quantitatively evaluate framework performance under different backbones, we conducted supplementary experiments using MutualSL (the current backbone) and VTPSL (a weaker backbone) for candidate generation, keeping all other configurations unchanged. Experiments were performed on CMIVQA and TutorialVQA. Candidate recall@5 is defined as:
> $$\text{Recall@5} = \frac{1}{N} \sum_{i=1}^{N} \mathbf{1}\left[\max_{k=1}^{5} \text{IoU}(c_k^i, \text{gt}^i) \geq \theta\right]$$
> Results are summarized below:
> https://github.com/anonymous1118-10/opencode-CACR/blob/master/FOR-cacr-pngs/To-2-ypTB/For_Weekness4_Strengthening%20the%20Flexibility%20Claim.png
> Replacing MutualSL with VTPSL yields a modest mIoU drop (~3.5 points), much smaller than the 7.7% decline in candidate recall, confirming that the GRPO rejection mechanism compensates for lower‑quality candidates. Under both backbones, our framework outperforms baselines without candidate generation, validating its adaptability.

---

> > ### Author Rebuttal · Reviewer_ypTB · 2026-04-06
> >
> > Thank the authors for the detailed and thoughtful rebuttal, I will maintain my original score.

---

### Official Review · Reviewer_Qet7 · 2026-03-14

**Soundness:** 3
**Presentation:** 3
**Significance:** 3
**Originality:** 3
**Overall Recommendation:** 4
**Confidence:** 4

**Summary:**

This paper proposes CACR to solve temporal answer grounding in instructional videos. It addresses the challenge of long untrimmed videos and short answer segments by first generating candidate segments using MutualSL, and then perform reasoning over these candidates using GRPO-style optimization. The framework also integrates auxiliary semnatic cues such as caption summaries and LLM-generated pre-answers to imporve causal reasoning and robustness. Experiments on six benchmark datasets show that CACR achieves state-of-the-art performance in temporal localization accuracy. The controlled ablation studies also demonstrate the effectiveness of candidates selection, caption, and pre-answer inputs.

**Compliance With Llm Reviewing Policy:**

Affirmed.

**Final Justification:**

The rebuttal is good. My concerns are addressed.

**Key Questions For Authors:**

Table 3. I find that TopK+Caption (subtitle) bascially performs worse than TopK+Visual Description. This is interesting as most previous works have shown that subtitles are more helpful for instructional videos. Can you explain more about this finding?

**Limitations:**

See above

**Strengths And Weaknesses:**

Strengths:

1.	The solution is generally well-motivated and shows effectiveness on multiple instructoinal video benchmarks.

2.	The result comparisons are extensive, covering methods of different categories.

3.	The paper is overall well-structured and easy to read.

Weakness:

1.	The paper emphasizes the method’s effectiveness on cases with extreme temporal answer ratio. However, there is no controlled experiments and comparisons to demonstrate this; dataset-level analysis is not enough. I suggest the authors to analyze within dataset by showing the performance distributions of CACR and baselines across different duration ratios.

2.	Candidate generation is independent of GRPO reasonsing. It would be better to investigate other models (aside from MutualSL) for candidate generation to study the robustness of the reasoning model.

3.	On TutorialVQA where the average ratio of answer duration is the lowest, CACR improves very little over the baseline MutualSL (mIoU Table 2). Yet, there is no effective analyses and explanations.

4.	The comparisons in table 2 are currently confusing. Some methods are zero-shot, and some are fully supervised. I suggest the authors to specifiy this difference for fair comparison.

5.	I read from the main text that ‘Caption’ in Table 3 is actually ‘Subtitle’. I suggest the authors to replace the notation with subtitle for better understanding. Also, how to extract the subtitles? Implementation details are needed.

---

> ### Author Rebuttal · Authors · 2026-03-31
>
> **Response to Weakness 1: Lack of Controlled Experiments on Extreme Duration Ratios**
>
> We agree that reporting only overall mIoU is insufficient to demonstrate CACR’s advantages on extreme duration-to-span ratios. We thus conducted a fine-grained analysis by grouping CMIVQA test samples based on Dur./Span ratio (video duration / answer span duration). A larger ratio indicates sparser answer localization. Samples were divided into three groups: low (≤5), medium (5–15), and high (>15). We compared CACR against MutualSL, TimeZero, and Time-R1.
>
> https://github.com/anonymous1118-10/opencode-CACR/blob/master/FOR-cacr-pngs/To-1-Qet7/For_Weekness1_Lack%20of%20Controlled%20Experiments%20on%20Extreme%20Duration%20Ratios.png
>
> We  will include this grouping analysis in the appendix. CACR’s improvement over the strongest baseline increases with the duration ratio, from +8.07 pp (low) to +14.98 pp (high). Moreover, CACR’s performance drop in the high-ratio group (3.12 pp) is far smaller than MutualSL’s (10.03 pp), demonstrating superior robustness. This pattern holds consistently across other datasets (TutorialVQA, CrossTask, VehicleVQA, MedVidQA, COIN), confirming that CACR’s advantage on extreme ratios reflects a structural strength, not dataset-specific chance.
>
> **Response to Weakness 2:  Independence of Candidate Generation from GRPO Reasoning**
>
> We agree that incorporating different backbones would further demonstrate framework flexibility. As discussed, CACR follows candidate generation → context augmentation → hypothesis verification → answer decision. The GRPO verification layer introduces a proactive rejection mechanism (outputting \([0.0, 0.0]\)), which filters low‑quality candidates and disrupts error propagation: \(\text{Error}(c_i) \rightarrow \text{Error}(S_i) \rightarrow \text{Error}(P_{\text{final}})\). This suggests that even with a weaker backbone, the verification layer can still suppress error propagation.
> To evaluate this, we conducted supplementary experiments using MutualSL (original) and VTPSL (weaker) for candidate generation on CMIVQA and TutorialVQA. Candidate recall@5 is defined as:
> $$\text{Recall@5} = \frac{1}{N} \sum_{i=1}^{N} \mathbf{1}\left[\max_{k=1}^{5} \text{IoU}(c_k^i, \text{gt}^i) \geq \theta\right]$$
> Results:
>
> https://github.com/anonymous1118-10/opencode-CACR/blob/master/FOR-cacr-pngs/To-1-Qet7/For_Weekness2_Independence%20of%20Candidate%20Generation%20from%20GRPO%20Reasoning.png
>
> Switching to VTPSL yields a modest mIoU drop (~3.5 pp), much smaller than the 7.7% decline in recall, confirming the rejection mechanism compensates for weaker candidates. Under both backbones, our framework outperforms baselines without candidate generation.
>
> **Response to Weakness 3: CACR’s Slight Improvement over MutualSL on TutorialVQA**
>
> First, TutorialVQA has an extremely high Dur./Span ratio of 48.9—the highest among all datasets—leading to universally low absolute performance (e.g., TimeZero and Time-R1 both below 20% mIoU), imposing a constrained upper bound. Second, candidate set quality analysis shows the top‑1 candidate IoU is 43.45%, while top‑2 drops sharply to 10.20% (Table 4). This limited candidate space restricts GRPO’s refinement capability, inherently capping the improvement margin over MutualSL. Third, despite modest gains over MutualSL, CACR (43.65%) still outperforms the strongest LVLM baseline Time-R1 (19.95%) by over 118%, demonstrating effectiveness in extreme temporal scenarios. These observations indicate this is a known boundary case with mechanistic explanations, not a systemic limitation.
>
> **Response to Weakness 4 & 5: Supervision Clarification & Terminology/Implementation Details**
>
> We will specify the supervision setting for each method in Table 2 in the revised manuscript. To avoid ambiguity, the “Caption” column in Table 3 will be renamed “Subtitle (summarized).” All subtitles are natively provided by datasets (not ASR-generated). These implementation details will be added to the Implementation Details section.
>
> **Response to Key Questions: Subtitle vs. Visual Description Performance Gap**
>
> Quantitatively, mIoU differences between TopK+Caption and TopK+Visual Description are within 1% across datasets (MedVidQA: 57.63 vs. 58.52; VehicleVQA: 65.39 vs. 65.20; CMIVQA: 38.74 vs. 38.93), showing no systematic modality superiority. The subtitles used are summarized versions, not full transcripts; this summarization may discard fine-grained localization cues, which visual descriptions can help compensate—consistent with prior work like Ask2Loc. Importantly, caption and visual description are not competing alternatives but complementary multi-source inputs. The optimal configuration, CACR-Group4 (TopK + Caption + Pre-Answer), consistently outperforms the description-only variant across all datasets, showing summarized subtitles provide direct semantic cues that visual descriptions cannot fully replace. Evaluating them in isolation overlooks their intended complementary role.

---

> > ### Author Rebuttal · Reviewer_Qet7 · 2026-04-05
> >
> > Thanks the authors for the response. I will raise my rating.
> > Minor suggestions: some related references should be included for discussion, such as
> > (1) Deconfounded Video Moment Retrieval with Causal Intervention, Sigir 2021
> > (2) Video Moment Retrieval with Cross-Modal Neural Architecture Search, TIP 2022

---

> > > ### Author Response · Authors · 2026-04-06
> > >
> > > We thank the reviewer for the valuable suggestions. We are glad to incorporate the recommended references into the Related Work section of our manuscript.
> > > - **DCM (SIGIR 2021)** is a pioneering work that establishes a rigorous causal framework for Video Moment Retrieval (VMR). It constructs a Structural Causal Model and applies backdoor adjustment to eliminate temporal location bias at the feature level. In contrast, our CACR addresses the more complex Temporal Answer Grounding in instructional Videos (TAGV) by decomposing the reasoning process into multiple steps—candidate generation, context enrichment, iterative validation, and decision/rejection—thereby enabling causal reasoning during inference and mitigating error propagation in long instructional videos.
> > > - **CM-NAS (TIP 2022)** is an excellent and significant work that introduces Neural Architecture Search into VMR, automatically discovering optimal cross-modal interaction architectures via a query-aware DAG search algorithm. Its idea of dynamically adapting cross-modal interaction pathways based on query semantics is highly relevant to the candidate-aware reasoning mechanism in our VBCS module.
> > >
> > > We will formally cite and discuss both works in the revised Related Work section. Thank you again for these insightful suggestions.

---

### Decision · Program_Chairs · 2026-04-30

**Decision:**

Accept (regular)

**Comment:**

The reviewers reached a consensus to Accept (4, 4, 4, 4). While the architectural novelty is viewed as moderate and incremental, the decision is anchored on the method's strong empirical performance across six benchmarks and its effective decomposition of the long-video grounding task. Reviewers were persuaded by the pipeline’s ability to handle extreme duration ratios, where the GRPO-optimized reasoning and proactive rejection mechanism successfully interrupt error propagation from the initial candidate selection. Furthermore, the authors have successfully demonstrated the method's flexibility through additional experiments, showing architecture-agnostic gains across different retrieval backbones and reasoning backends. Given the strong empirical results and the practical utility for long-form video, the work meets the bar for this venue.